# TROPICAL EXPRESSIVITY OF NEURAL NETWORKS

## ABSTRACT

We propose an algebraic geometric framework to study the expressivity of piece-wise linear activation neural networks. A particular quantity of neural networks that has been actively studied is the number of linear regions, which gives a quantification of the information capacity of the architecture. To study and evaluate information capacity and expressivity, we work in the setting of tropical geometry—a combinatorial and polyhedral variant of algebraic geometry—where there are known connections between tropical rational maps and feedforward neural networks. Our work builds on and expands this connection to capitalize on the rich theory of tropical geometry to characterize and study various architectural aspects of neural networks. Our contributions are threefold: we provide a novel tropical geometric approach to selecting sampling domains among linear regions; an algebraic result allowing for a guided restriction of the sampling domain for network architectures with symmetries; and a new open source OSCAR library to analyze neural networks symbolically using their tropical representations, where we present a new algorithm that computes the exact number of their linear regions. We provide a comprehensive set of proof-of-concept numerical experiments demonstrating the breadth of neural network architectures to which tropical geometric theory can be applied to reveal insights on expressivity characteristics of a network. Our work provides the foundations for the adaptation of both theory and existing software from computational tropical geometry and symbolic computation to neural networks and deep learning.

## 1   INTRODUCTION

Deep learning has become the undisputed state-of-the-art for data analysis and has wide-reaching prominence in many fields of computer science, despite still being based on a limited theoretical foundation. Developing such a foundation to better understand the unparalleled success of deep neural networks is one of the most active areas of research in modern statistical learning theory; with attempts at characterising the *expressivity* – the space of representable functions – of deep neural networks being one of the most important approaches to this Raghu et al. (2017). Our focus is on deep neural networks with piecewise linear activations since their expressivity has been extensively studied using linear regions (e.g., Pascanu et al., 2013; Montúfar et al., 2014; Arora et al., 2016; Hanin & Rolnick, 2019; Xiong et al., 2020; Goujon et al., 2024; Montúfar et al., 2022).

*Tropical geometry* is a reinterpretation of algebraic geometry that features piecewise linear and poly-hedral constructions (see Appendix A.3), where combinatorics naturally comes into play Mikhalkin & Rau (e.g., 2009); Speyer & Sturmfels (e.g., 2009); Maclagan & Sturmfels (e.g., 2021). Therefore, tropical geometry is a natural framework for studying the linear regions in a neural network. The intersection of deep learning theory and tropical geometry is a relatively new area of research with great potential towards the ultimate goal of understanding how and why deep neural networks perform so well. In this paper, we expand the connection between deep learning theory and tropical geometry by studying the linear regions of neural networks through a tropical lens. In doing so, we introduce a novel perspective on studying neural network expressivity.

**Related Work.**   Tropical geometry has emerged as a powerful tool for analyzing deep neural networks with piecewise linear activation functions, such as rectified linear units (ReLUs) and maxout units. Zhang et al. (2018) first established that neural networks can be represented by tropical rational functions, enabling the use of tropical techniques to study their properties. They also showed

that the decision boundary of a deep ReLU network is contained in a tropical hypersurface. (For more details, see Appendix A.3.) Concurrently, Charisopoulos & Maragos (2018b) demonstrated that the maxout activation function fits input data using a tropical polynomial. These initial works focused on neural networks with Euclidean input domains. Yoshida et al. (2023) later extended this approach to incorporate the tropical projective torus as an input domain, broadening the applicability of tropical methods. Recently, Pasque et al. (2024) leveraged tropical geometry to construct convolutional neural networks with improved robustness against adversarial attacks, demonstrating the practical value of this theoretical framework. This growing body of research highlights the potential of tropical geometry to enhance our understanding and design of neural networks.

Measures of quantifying neural network expressivity are important for facilitating the theoretical and empirical investigation of neural network properties. Neural networks with piecewise linear activations compute piecewise linear functions on areas of the input space referred to as the network's *linear regions*. The *maximum* number of distinct linear regions a class of neural networks can instantiate provides an appropriate and quantifiable measure of its expressivity (e.g., Montúfar et al., 2014). Subsequent research has thus worked towards enumerating the number of linear regions in a given neural network since this can provide a measure of how complex the function represented by the neural network is. Using mixed-integer programming, Serra et al. (2018) provides an exact enumeration of the number of linear regions in a bounded subset of the input domain. For unbounded domains, Serra & Ramalingam (2018) give an analytic upper bound on the maximum number of linear regions of a neural network along with a probabilistic lower bound, and Charisopoulos & Maragos (2018a) provide a probabilistic method for estimating the linear regions of a one-layer network, together with some analytic bounds for various architectures. In recognition that many of these approaches are intractable at large scales, linear region enumeration is often done approximately through numerical sampling Goujon et al. (2024).

However, the work that is most related to ours are those that exactly characterise the linear regions, beyond just enumerating them, in a computationally tractable manner. For instance, Humayun et al. (2023) leverages spline theory to do so in bounded two-dimensional subspaces of the input domain. Similarly, Masden (2022) characterises the polyhedral complex associated to the arrangement of the linear regions. Our work differs from these in that we obtain an exact geometric characterisation of the individual regions in the unbounded input domain.

**Contributions.** In this paper, we establish novel algebraic and geometric tools to obtain previously inaccessible geometric insights into the linear regions of a given neural network beyond just their enumeration; enhancing the theoretical and empirical (exact or approximate) study of a neural network's linear regions. The main contributions of our work are the following:

- We provide a global geometric characterization of the arrangement of a neural network's linear regions: current numerical estimation of the number of linear regions is typically carried out by random sampling in an arbitrarily bounded region of the input space, potentially causing some linear regions of a neural network to be missed and resulting in an inaccurate information capacity measure. We propose an *effective sampling domain* as a ball of radius $R$ that hits all of the linear regions of a given neural network. We compute bounds for the radius $R$ based on a combinatorial invariant known as the *Hoffman constant*, giving a guarantee on the positioning of the linear regions of a neural network.

- We exploit further geometric insight into the arrangement of linear regions of a neural network to gain dramatic computational efficiency in the numerical estimation of linear regions: when networks exhibit invariance under symmetry, we can restrict the sampling domain to a *fundamental domain* of the group action and thus reduce the number of samples required. We experimentally demonstrate that sampling from the fundamental domain provides an accurate estimate of the number of linear regions with a fraction of the compute requirements.

- We provide an open source library integrated into the Open Source Computer Algebra Research (OSCAR) system (OSCAR) which converts arbitrary neural networks into algebraic symbolic objects, to facilitate the *exact* enumeration of the linear regions of neural networks. Our library opens the door for the extensive theory and existing software on symbolic computation and computational tropical geometry to be used to study neural net-

works beyond their linear regions. In particular, we propose an alternative measure of neural network complexity, the *monomial count*.

The remainder of this paper is organized as follows. We devote a section to each of the contributions listed above—Sections 2, 3, and 4, respectively—in which we present our theoretical contributions and numerical experiments. We close the paper with a discussion on limitations of our work and directions for future research in Section 5. The Appendix provides the necessary technical background on tropical geometry and its connection to neural networks as well as all proofs.

## 2 BOUNDED DOMAIN SELECTION USING A HOFFMAN CONSTANT

Enumerating the number of linear regions of a given neural network is inherently challenging due to its combinatorial nature. From an applications perspective, it is common to restrict to some bounded input domain $X \subseteq \mathbb{R}^n$ such as $[-R, R]^n$ or $[0, R]^n$, such that the number of linear regions in $X$ can be either estimated by computing the Jacobians of the network at sample points (see Appendix C) or computed exactly using mixed-integer programming (e.g., Serra et al., 2018). However, these approaches are limited from both a theoretical and empirical perspective, since, one cannot guarantee that $X$ provides information about all the linear regions. In this section, we try to address the discrepancy by inferring the unbounded case from the bounded case. More specifically, we provide a method to determine the radius $R$ of a ball centred at some point $x$, which intersects every linear region.

Our approach proceeds from the recollection that neural networks can be formulated as tropical Puiseux rational maps (see Appendix A). Thus, characterizing $R$ for neural networks is equivalent to characterizing $R$ for tropical Puiseux rational maps. The linear regions of tropical Puiseux polynomials are made up of polyhedra, allowing us to connect the value $R$ to a combinatorial invariant: the *Hoffman constant*. We extend the definition of the Hoffman constant to tropical rational maps and use it to derive bounds on $R$.

### 2.1 NEURAL HOFFMAN CONSTANTS

**The Hoffman Constant for Polyhedra.** A polyhedron can be constructed through a series of linear constraints. Intuitively, the Hoffman constant of a polyhedron captures the stability of points that satisfy those constraints in terms of distance. That is, if the Hoffman constant is large, then the polyhedron has near-contradictory constraints such that points narrowly violating these constraints are a relatively large distance away from the polyhedron. We formalise this using polyhedral geometry which we introduce, along with relevant notation in Appendix A.

Let $A$ be an $m \times n$ matrix with real entries. Then for any $b \in \mathbb{R}^m$ such that $P(A, b)$ is non-empty, let
$$d(u, P(A, b)) = \min\{\|u - x\| : x \in P(A, b)\}$$
denote the distance of a point $u \in \mathbb{R}^n$ to the polyhedron, measured under an arbitrary norm $\|\cdot\|$ on $\mathbb{R}^n$. Then there exists a constant $H(A)$ only depending on $A$ such that
$$d(u, P(A, b)) \leq H(A) \|(Au - b)_+\|, \tag{1}$$
where $x_+ = \max(x, 0)$ is applied coordinate-wise (Hoffman, 2003). $H(A)$ is the *Hoffman constant* of $A$. Intuitively, $\|(Au - b)_+\|$ can be thought of as quantifying the degree to which $u$ violates the constraints of the polyhedron, and $H(A)$ can be thought of as determining the extent to which this affects the distance of $u$ to the polyhedron.

**The Hoffman Constant for Tropical Polynomials.** Let $f : \mathbb{R}^n \to \mathbb{R}$ be a tropical Puiseux polynomial and let $\mathcal{U} = \{U_1, \ldots, U_m\}$ be the set of linear regions of $f$. Say $f(x) = a_{i1}x_1 + \ldots + a_{in}x_n + b_i$ occurs on the region $U_i$, so that $A = [a_{ij}]_{m \times n}$ is the matrix of exponents in the algebraic expression of $f$. The linear region $U_i$ is defined by the inequalities
$$a_{i1}x_1 + \cdots + a_{in}x_n + b_i \geq a_{j1}x_1 + \cdots + a_{jn}x_n + b_j, \quad \forall\, j = 1, 2, \cdots, m \tag{2}$$
which can be written in matrix form as
$$(A - \mathbf{1}a_i)x \leq b_i\mathbf{1} - b, \tag{3}$$

where $\mathbf{1}$ is an all-1 column vector; $a_i$ is the $i$th row vector of $A$; and $b$ is a column vector of all $b_i$. Denote $\widetilde{A}_{U_i} := A - \mathbf{1}a_i$ and $\widetilde{b}_{U_i} := b_i\mathbf{1} - b$. Then the linear region $U_i$ is captured by the linear system of inequalities $\widetilde{A}_{U_i}x \leq \widetilde{b}_{U_i}$.

**Definition 2.1.** Let $f : \mathbb{R}^n \to \mathbb{R}$ be a tropical Puiseux polynomial. The *Hoffman constant of $f$* is

$$H(f) = \max_{U_i \in \mathcal{U}} H\big(\widetilde{A}_{U_i}\big).$$

Thus, the Hoffman constant of $f$ can be thought of as the maximum *instability* encountered by points trying to satisfy the constraints of $f$'s linear regions.

**The Hoffman Constant for Tropical Rational Maps.** Care needs to be taken here as we can no longer assume that all linear regions are defined by systems of linear inequalities, since such maps can admit non-convex linear regions. To bypass this difficulty, we consider convex refinements of linear regions induced by intersections of linear regions of tropical polynomials.

**Definition 2.2.** Let $p \oslash q$ be a difference of two tropical Puiseux polynomials. Let $A$ (respectively $A'$) be the $m_p \times n$ (respectively $m_q \times n$) matrix of exponents for $p$ (respectively $q$), and $a_i$ (respectively $a_i'$) the $i$th row vector of $A$ (respectively $A'$). The Hoffman constant of $p \oslash q$ is

$$H(p \oslash q) := \max \left\{ H\left( \begin{bmatrix} A \\ A' \end{bmatrix} - \mathbf{1} \begin{bmatrix} a_i \\ a_j' \end{bmatrix} \right) : i = 1, \cdots, m_p; \ j = 1, \cdots, m_q \right\}. \tag{4}$$

Let $f$ be a tropical Puiseux rational map. Then the *Hoffman constant of $f$* is defined as the minimal Hoffman constant of $H(p \oslash q)$ over all possible expressions of $f = p \oslash q$.

As every neural network can be represented by a tropical Puiseux rational map, Definition 2.2 gives a notion of Hoffman constants for neural networks.

Since the Hoffman constants contributing to the calculation of $H(p \oslash q)$ represent the Hoffman constants of polyhedra obtained by intersecting the $i$th linear region of $p$ and the $j$th linear region of $q$, $H(p \oslash q)$ can be similarly interpreted as maximum instability encountered by points trying to satisfy the constraints of $p \oslash q$'s linear regions.

## 2.2 THE MINIMAL EFFECTIVE RADIUS

We can now utilise these ideas to grapple with the effective radius. For a neural network whose tropical Puiseux rational map is $f : \mathbb{R}^n \to \mathbb{R}$, let $\mathcal{U} = \{U_1, \ldots, U_m\}$ be the collection of its linear regions. For any $x \in \mathbb{R}^n$, define the *minimal effective radius* of $f$ at $x$ as

$$R_f(x) := \min \{r : B(x, r) \cap U \neq \varnothing, \ U \in \mathcal{U}\}$$

where $B(x, r)$ is the ball of radius $r$ centered at $x$. That is, $R_f(x)$ is the minimal radius such that the ball $B(x, r)$ intersects all linear regions.

**Lemma 2.3.** *Let $f$ be a tropical Puiseux polynomial and $x \in \mathbb{R}^n$ be any point. Then*

$$R_f(x) \leq H(f) \max_{U \in \mathcal{U}} \big( \big\| (\widetilde{A}_U x - \widetilde{b}_U)_+ \big\| \big). \tag{5}$$

In particular, we are interested in the case when $\mathbb{R}^m$ and $\mathbb{R}^n$ are equipped with the $\infty$-norm, where the minimal effective radius can be related to the Hoffman constant and function value of $f = p \oslash q$. For a tropical Puiseux polynomial $p(x) = \max_{1 \leq i \leq m_p} \{a_i x + b_i\}$, we set $\check{p}(x) = \min_{1 \leq j \leq m_p} \{a_j x + b_j\}$ to be its min-conjugate.

**Proposition 2.4.** *Let $f = p \oslash q$ be a tropical Puiseux rational map. For any $x \in \mathbb{R}^n$, we have*

$$R_f(x) \leq H(p \oslash q) \max\{p(x) - \check{p}(x), \ q(x) - \check{q}(x)\}. \tag{6}$$

We have thus related the minimal effective radius of a tropical Puiseux rational map to its Hoffman constant.

We started this section motivated to obtain a bound on the effective radius to provide guarantees for the numerical sampling of a neural network's linear region. Although we provide algorithms to compute the value of the Hoffman constant exactly, along with lower and upper bounds — refer to Section E – due to its combinatorial nature it seems largely intractable for practical purposes. Despite this, we have still provided a theoretical connection between the bounded and unbounded case, and we can leverage our intuition of the Hoffman constant to gain further insights. More specifically, Proposition 2.4 encourages us to promote the construction of stable polyhedra around points of interest (training data) or known complexities in the input domain; this would attract the linear regions of the neural network to these regions, improving the network's expressivity in these regions. We leave it for future work to identify ways this could be achieved, say through initialisation schemes, training methodologies or architectural choices.

## 3    SYMMETRY AND THE FUNDAMENTAL DOMAIN

Here we continue to leverage the geometric characterisation of linear regions as polyhedra to tangibly optimise the empirical enumeration of neural network linear regions.

### 3.1    THE LINEAR STRUCTURE OF INVARIANT NETWORKS

The notions of *invariance* and *equivariance* under symmetries are central to *geometric deep learning* (Bronstein et al., 2021), which leverages the inherent symmetries of data so that models generalize more effectively (Sannai & Imaizumi, 2019). In our setting, symmetries in a neural network induce symmetries in the linear structure of the network (see Figure 1), which we can exploit for computational gains.

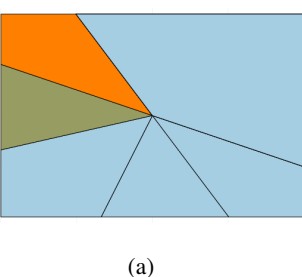
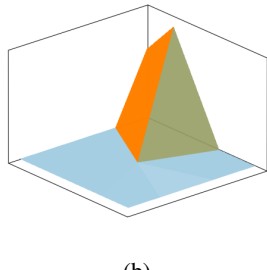

(a)                                                              (b)

Figure 1: A visualisation of the linear regions, 1a, and the corresponding linear maps, 1b, of a randomly initialized permutation invariant neural network. The regions are colour-coded according to which linear maps operate on these regions.

**Definition 3.1.** Let $f : \mathbb{R}^n \to \mathbb{R}$ be a function and let $G$ be a group acting on the domain $\mathbb{R}^n$. $f$ is said to be *invariant* under the group action of $G$ (or $G$-invariant) if for any $g \in G$, $f \circ g = f$.

It seems reasonable to incorporate the effect of the group action into constructing the sampling domain for empirically enumerating the linear regions.

**Definition 3.2.** Let $G$ be a group acting on $\mathbb{R}^n$. A subset $\Delta \subseteq \mathbb{R}^n$ is a *fundamental domain* if it satisfies the two following conditions: (i) $\mathbb{R}^n = \bigcup_{g \in G} g \cdot \Delta$; and (ii) $g \cdot \text{int}(\Delta) \cap h \cdot \text{int}(\Delta) = \varnothing$ for all $g, h \in G$ with $g \neq h$.

Even though Definitions 3.1 and 3.2 apply to any group $G$, we will consider $G$ to be finite such that the fundamental domain induces a periodic tiling of $\mathbb{R}^n$ by acting on $\Delta$; which is very useful in the context of numerical sampling since it means we can sample from a smaller subset of the input domain with a guarantee to find all the linear regions in the limit. The upshot is that we can use far fewer samples while maintaining the same density of points.

**Theorem 3.3.** *Let $f : \mathbb{R}^n \to \mathbb{R}$ be a tropical rational map invariant under the group action $G$, where $G$ is finite. Let $\Delta \subseteq \mathbb{R}^n$ be a fundamental domain of $G$. Suppose $\mathcal{U}$ is the set of linear regions of $f$. Define the sets*

$$\mathcal{U}_c := \{A \in \mathcal{U} : A \subseteq \Delta\} \quad and \quad \mathcal{U}_e := \{A \in \mathcal{U} : A \cap \Delta \neq \varnothing\}.$$

*Then*

$$|G||\mathcal{U}_c| \leq |\mathcal{U}| \leq |G||\mathcal{U}_c| + \sum_{A \in \mathcal{U}_e \setminus \mathcal{U}_c} \frac{|G|}{|G_A|}.$$

*where $|G_A|$ is the size of the stabilizer of $A$.*

Theorem 3.3 gives us a method for estimating the total number of linear regions from sampling in the fundamental domain using *multiplicity*, which we discuss next.

## 3.2 ESTIMATING LINEAR REGIONS USING THE FUNDAMENTAL DOMAIN

We now demonstrate the performance improvements in counting linear regions gained by exploiting symmetry in the network architecture with a study of permutation invariant neural networks inspired by DeepSets (Zaheer et al., 2017). Our numerical sampling approach is detailed in Appendix C and inspired by recent work in this area (Goujon et al., 2024). Here, we focus on a specific sampling method for estimating the number of linear regions for illustrative purposes, but we emphasize that our approach based on Theorem 3.3 is readily adaptable to any method for determining the number of linear regions on a bounded domain.

A permutation invariant network is one that is invariant under the action of $S_n$ on coordinates (see Appendix A.4). Intuitively – and theoretically as a consequence of Lemma A.19 – this action has fundamental domain

$$\Delta = \{(x_1, \ldots, x_n) : x_1 \geq x_2 \geq \ldots \geq x_n\},$$

since it is possible to map any point to $\Delta$ using a permutation, and $\Delta$ forms an $n!$ piece tiling of $\mathbb{R}^n$ under the $S_n$-action. Thus, despite restricting sampling to $\Delta$, we can still effectively gain information about linear regions outside $\Delta$.

Our method of numerical sampling characterises linear regions with Jacobians of the neural network with respect to the inputs, meaning linear regions are identified by $n$-dimensional real vectors. Thus, to estimate the number of linear regions of this neural network, we need to address the multiplicities of these vectors.

**Lemma 3.4.** *Let $f : \mathbb{R}^n \to \mathbb{R}$ be a permutation invariant neural network as given by equation 9. Let $J$ be the Jacobian of the neural network at the point $x \in \mathbb{R}^n$. Then $f$ has at most*

$$\mathrm{mult}(J) = \frac{n!}{\prod_{c \in C(J)} c!}$$

*distinct linear regions with the corresponding linear map having Jacobian $J$, where $C(J)$ gives the counts of each of the elements of $J$.*

By Lemma 3.4, the number of linear regions of the neural network can be estimated by $\sum_{J \in \mathcal{J}} \frac{n!}{\prod_{c \in C(J)} c!}$, where $\mathcal{J}$ are the Jacobians of the linear regions in $\Delta$ computed by Algorithm 3. Consequently, we can estimate the number of linear regions of the neural network while reducing the number of point samples by a factor of $n!$. This provides a dramatic gain in computational efficiency via an upper bound rather than an exact number; see Appendix C.3 for empirical demonstrations.

## 4 SYMBOLIC NEURAL NETWORKS

In the previous sections we understood how our algebraic characterisation of neural networks can be used to generate theoretical insights into the global geometry of its linear regions.

In this section, we present our threefold contribution of symbolic tools for neural networks, which characterise the local geometry of linear regions as well as providing additional insights. We present

this as a new Julia library integrated into the OSCAR system: (i) an algorithm to determine algebraic representations of the linear regions of arbitrary tropical Puiseux rational functions; (ii) methods for computing the tropical representations of neural networks and simplifying them; and (iii) A new algebraic measure of complexity for neural networks, *monomial complexity*.

The combinations of these tools allow us to compute algebraic representations of the linear regions of *arbitrary* neural networks, and enumerate them *exactly*.

The Julia library forming a part of our symbolic contribution can be found in the following anonymized repository:
https://anonymous.4open.science/r/tropical-expressivity/README.md,

## 4.1 Linear Regions of Tropical Puiseux Rational Functions

**Overview of the Algorithm.** We start by sketching our algorithm for determining the linear regions of tropical Puiseux rational functions. A more precise formulation is given in the Appendix B ; see Algorithm 1, together with a proof of correctness (Theorem F.1), and a Julia implementation.

When viewed as a function on the real numbers, a tropical Puiseux rational function $f = p \oslash q$ is simply the difference of two max terms: the numerator and the denominator. In particular, $f$ is linear on a region $R \subset \mathbb{R}^n$ whenever $p$ and $q$ are linear on $R$. This indicates that we should be able to determine the linear regions of $f$ once we know those of $p$ and $q$. More precisely, notice that if $U_1, \ldots, U_s$ are the linear regions of $p$ and $V_1, \ldots, V_t$ are the linear regions of $q$, then $f$ is linear on each of the intersections $U_i \cap V_j$, and these intersections cover the input space $\mathbb{R}^n$. However, we cannot conclude that the linear regions of $f$ are given by this collection of intersections, as the following issues may arise: some intersections may be empty or have dimension less than $n$; and some intersections may "glue" together to form a larger linear region of $f$ (see Appendix H for examples of these phenomena). Whether or not these arise usually depends on the $U_i$'s and $V_j$'s and has to do with the combinatorics of the arrangement of these objects in $\mathbb{R}^n$. After filtering out empty and lower-dimensional regions, and determining which intersections glue together, we obtain a list of regions in $\mathbb{R}^n$ (polyhedra or unions of polyhedra), which correspond to the linear regions of $f$.

**Combinatorics of Polyhedral Arrangements.** In order for our tropical linear region algorithm to be implementable, we need a way of computationally determining the combinatorics of the arrangement of the $U_i$'s and $V_j$'s. The key here is the standard fact from tropical geometry that the linear regions of a tropical (Puiseux) polynomial are polyhedra whose defining inequalities can be determined from the coefficients and exponents of the polynomial (see Appendix A.3 for more detail). Hence, we are left to deal with the combinatorics of polyhedral arrangements.

From Lemma A.4, we have that the intersections of the linear regions of $p$ and $q$ are also polyhedra, and thus determining the non-emptiness or dimension of such objects are well-understood problems that can be solved using linear programming. This means that we can (computably!) detect when some intersections may be empty or have dimension less than $n$ using polyhedral geometric tools.

Next, to deal with gluing intersections, we can once again reduce to a problem of intersections of polyhedra. Let us denote by $L_i$ the linear map representing $p$ on $P_i$ and similarly, we write $M_j$ for the linear map representing $q$ on $Q_j$. Then gluing intersections may arise when there exist tuples of indices $(i, j, k, \ell)$ that satisfy the following set of conditions, $(\star)$:

(i) The intersection $(U_i \cap V_j) \cap (U_k \cap V_\ell)$ is non-empty;

(ii) $f$ is represented by the same linear map on $U_i \cap V_j$ and on $U_k \cap V_\ell$; and

(iii) $\dim(U_i \cap V_j) = \dim(U_k \cap V_\ell) = n$.

Notice that $f$ is represented by the same linear map on $U_i \cap V_j$ and on $U_k \cap V_\ell$ if and only if the equality of linear maps $L_i - M_j = L_k - M_\ell$ holds. Thus we can computably determine when such indices arise. For some fixed indices $i, j$ such that $\dim U_i \cap V_j = n$, two cases can arise:

(a) Either there are no pairs of indices $(k, \ell) \neq (i, j)$ such that $(i, j, k, \ell)$ satisfies $(\star)$; or

(b) There exist pairs of indices $(k, \ell) \neq (i, j)$ such that $(i, j, k, \ell)$ satisfies $(\star)$.

When (a) occurs, $U_i \cap V_j$ is a linear region of $f$. We now focus on (b): Set $\mathcal{I}$ to be the set of all pairs of indices $(k, \ell)$ such that $(i, j, k, \ell)$ satisfies $(\star)$ and $F = L_i - M_j$. Then, $f$ is represented by $F$ on the (possibly disconnected) region

$$\bigcup_{(k,\ell) \in \mathcal{I}} U_k \cap V_\ell,$$

and the linear regions where $f$ is represented by $F$ correspond to the connected components of this region. We can determine these computationally as unions of polyhedra by considering which pairwise intersections $(U_k \cap V_\ell) \cap (U_{k'} \cap V_{\ell'})$ are empty for $(k, \ell), (k', \ell') \in \mathcal{I}$.

Hence, we have shown that given a tropical rational map we can computably determine its linear regions. Therefore, to be able to determine a neural network's linear regions we just need to computationally realise it as a tropical rational, which we discuss in the next section.

## 4.2 Tropical Representations of Neural Networks

**Our Contribution.** Any neural network with integer weights can be viewed as the function $\mathbb{R}^n \to \mathbb{R}$ associated to a tropical rational function (Zhang et al., 2018). This fact is used by Brandenburg et al. (2024) as a theoretical tool, but to the best of our knowledge, this has not yet been implemented in practice for analyzing concrete neural networks. We fill this gap by leveraging the Julia package OSCAR to computationally realise this representation using the constructive proof provided in (Zhang et al., 2018), see Section A.3 for details.

However, non-zero tropical Puiseux rational maps (and polynomials) induce functions $\mathbb{R}^n \to \mathbb{R}$ which can be realised by different algebraic expressions, see 3 and 4 of Example A.14. In this sense, the algebraic expressions contain strictly more information than the corresponding function. Since for neural networks we are only concerned with the induced functions, it is natural to consider whether the tropical representation of these neural networks is optimal, in the sense of having a relatively few number of redundant monomials.

This observation extends to interpretability, where a goal is to find minimal expressions of neural networks. Tropical geometry has been used for finding such representations (Smyrnis et al., 2020; Smyrnis & Maragos, 2020), where the corresponding minimal representations have been studied in algebraic statistics (Tran & Wang, 2024). Our contribution brings a new perspective to expressivity as well as interpretability using polyhedral geometry.

**Pruning Tropical Expressions.** If $g = \bigoplus_{j=1}^m a_{\alpha_j} T^{\alpha_j}$ is a tropical Puiseux polynomial in $n$ variables, then we can associate to each monomial $a_{\alpha_i} T^{\alpha_i}$ a polytope $P_i \subset \mathbb{R}^n$ such that the maximum in the expression

$$g(x) = \max_{j \in \{1, \dots, m\}} \left\{ a_{\alpha_j} + \langle \alpha_j, x \rangle \right\} \tag{7}$$

is attained at the $i$th term precisely when $x \in P_i$ (see Appendix A.3 for further details). The following lemma gives a natural criterion for detecting which monomials are redundant – in the sense that they do not effect the induced function – based on the geometry of their associated polyhedra.

**Lemma 4.1.** *The $i$th monomial can be removed from the expression of $g$ without changing the corresponding function $\mathbb{R}^n \to \mathbb{R}$ if and only if $\dim P_i < n$.*

In particular, this gives us a computable way of simplifying tropical expressions – presented in Algorithm 2 – of neural networks and measuring their monomial complexity. Henceforth, we refer to the monomial count of a tropical rational map as the sum of the monomials in the numerator and denominator of its pruned representation.

## 4.3 Symbolic Experiments

We now demonstrate the breadth of our symbolic contribution via exploratory experiments. We aim to demonstrate the new possibilities that our approach opens up, rather than merely providing performance metrics.

Throughout, we restrict our explorations to low-dimensional input spaces and small architectures, due to the combinatorial nature of these computations. This is is in line with explorations of similar

methods. For instance, Serra et al. (2018); Hu et al. (2022); Masden (2022) analyse the linear regions of neural networks with no more than 2 layers of width at most 16 on bounded input domains. Our tools, however, consider the full input domain of these neural networks and provide an exact geometric characterization of the linear regions; opening up previously inaccessible avenues for analyzing the geometry of linear regions of networks. In Figure 7 we demonstrate how our tools can be used for the exact enumeration of linear regions, and its connection to the notion of monomial complexity.

**Volumes of Linear Regions Through Training.** Previously mentioned works that characterise exactly the linear regions do not facilitate geometrical computations such as computing the volumes of the individual regions. For instance, Humayun et al. (2023) computes the average volume of the regions within a bounded domain, whereas Masden (2022) extracts topological information of the decision boundary.

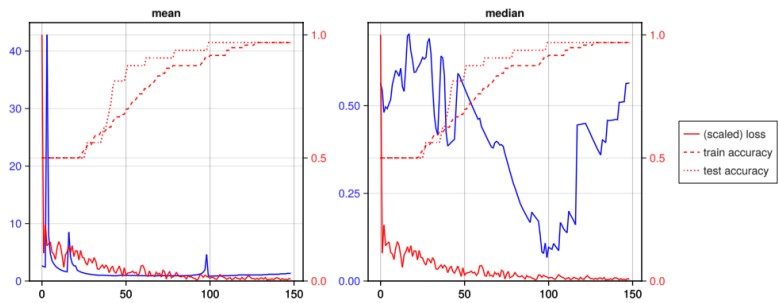

Figure 2: Statistics on the volumes of bounded linear regions in a neural network throughout training.

In Figure 2 we compute the exact volumes of the bounded linear regions through training. Spikes in the mean volumes indicate regions becoming unbounded. Interestingly we see a reduction in the median volume of the regions until the point where train and test accuracy saturate, after which the median volume increases.

**Redundant Monomials.** Our tools construct the tropical representation of a neural network through a standard procedure involving the weights of the network, which leads to a *native* tropical representation of the neural network. As discussed previously we can prune this representation of redundant monomials, a detailed investigation into the nature of these redundant monomials is beyond the scope of this work, although Appendix D outlines a basis for exploring the idea of monomial complexity.

Here we demonstrate the implementation of Algorithm 2 by pruning the native tropical representation of 10 randomly initialised neural networks of architectures $[2, k, 1]$ for $k \in \{4, 5, 6, 7, 8\}$ and $[2, k, 2, 1]$ for $k \in \{2, 3, 4, 5, 6\}$. Note that our choices of $k$ allow us to compare neural networks with the same number of hidden neurons.

From Figure 3, we observe that the pruning rates are relatively high, in particular for deeper neural networks. It will be left to future work to understand whether this is due to inefficiencies in the construction algorithm, or an implicit bias from our method of random initalisation. Moreover, Figure 3 demonstrates that depth provides exponentially more monomials than width, which concurs with existing literature Telgarsky (2016).

**At the MNIST Scale.** We now demonstrate the implementation of our tools on neural networks with a practical input domain. We are not looking to derive any specific insights, but rather demonstrate the potential utility of our tools. We train a neural network with a $[784, 4, 10]$ architecture on the MNIST dataset and achieve 85% accuracy on the train and test dataset. Using our tools we can obtain the neural network's tropical representation, which has 144 monomials. Moreover, we enumerate exactly its 9 linear regions, along with their polyhedral representations. In particular, we can deduce that 8 of the linear regions are on a single unbounded polyhedron; one linear region is a collection of several very small bounded polyhedra and one unbounded polyhedron.

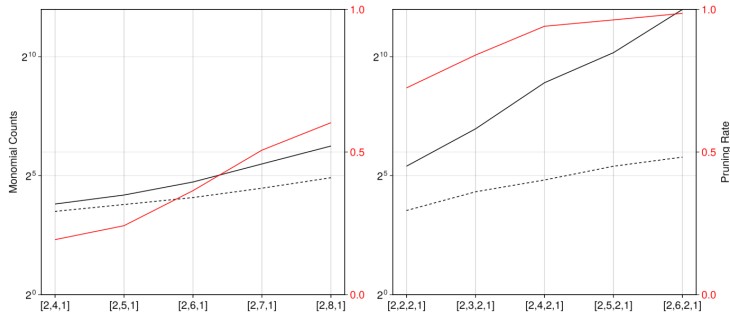

Figure 3: Solid black lines represent the number of monomials in the native representation and the dashed black lines correspond to the number of monomials in the pruned representation. The red line corresponds to the pruning rate. Notice how the $y$-axes for monomial counts are logarithmic and aligned. The $x$-axes are aligned such that relative points on the plots correspond to architectures with the same number of hidden neurons.

## 5  DISCUSSION: LIMITATIONS & DIRECTIONS FOR FUTURE RESEARCH

Our contributions offer theoretical and practical advancements in *tropical deep learning*, but are subject to some important limitations which in turn inspire directions for future research, which we now discuss.

**Experimental Considerations.**  Our methods have shown promising results for networks of moderate size, including those with input dimensions comparable to MNIST. However, as we scale to more complex architectures and higher dimensions, computational challenges persist. To further improve scalability, parallelization of our elementary computations could yield significant performance gains, as many of our algorithms involve repeating similar operations multiple times.

**Structural Considerations.**  Some of the problems we study are framed as combinatorial optimization problems, which are inherently challenging. For instance, computing the Hoffman constant, which is equivalent to the Stewart–Todd condition measure of a matrix, is known to be NP-hard in general cases (Peña et al., 2018; 2019). This challenge could be addressed by employing approximate algorithms or algorithms that provide upper bounds on the Hoffman constant, since these would be sufficient for our purposes and computationally more tractable.

Our introduction of a new algebraic measure of complexity provides fresh insights but also opens up new questions about its computational complexity and relationship to other complexity measures. The neural network pruning methods we have developed show promise in reducing model complexity while maintaining expressivity. However, further research is needed to fully understand the trade-offs between model size, expressivity, and performance across a wider range of architectures and tasks.

**Future Directions.**  These limitations inspire future work on both the practical and theoretical fronts. In practice, to achieve improved scalability, further studying and understanding where and how symbolic computation algorithms can be made more efficient, e.g., by parallelization or novel algorithmic approaches, would make our proposed methods more applicable to larger neural networks. Expanding our pruning methods to a broader range of architectures to investigate their impact on model performance in diverse tasks is a pathway to developing improved pruning techniques. Building on our initial empirical tests of theoretical expressivity results, a comprehensive empirical validation could help bridge the gap between theory and practice in neural network expressivity.

Theoretically, our tropical contributions have the potential to capture both expressivity and interpretability. Towards this end, a deeper exploration of our new algebraic expressivity measure, including its theoretical properties and practical implications, could yield valuable insights into neural network behaviour. Perhaps most importantly, ours is the first work to leverage tropical symbolic computation to perform experiments on deep neural networks. Fostering collaboration between these fields will lead to novel algorithms and insights that leverage the strengths of both areas.

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

## A  TECHNICAL BACKGROUND

In this section, we provide the necessary technical background on the mathematics of neural networks and tropical geometry for our contributions.

### A.1  NEURAL NETWORKS

**Definition A.1.** Given a function $\sigma : \mathbb{R} \to \mathbb{R}$, a *neural network with activation $\sigma$* is a function $f : \mathbb{R}^n \to \mathbb{R}^m$ of the form

$$\sigma \circ L_d \circ \cdots \circ \sigma \circ L_1$$

where $L_i : \mathbb{R}^{n_{i-1}} \to \mathbb{R}^{n_i}$ is an affine map and the function $\sigma$ is applied to vectors element-wise. The tuple of integers $[n_0, \ldots, n_d]$ is called the *architecture* of the neural network $f$.

Given such a neural network, we can always write the $L_i$ as $L_i(x) = A_i x + b_i$, with $A_i$ as a *weight matrix* and $b_i$ is a *bias vector* for the $i$th layer of the neural network; $\mathbb{R}^n$ is the *input domain* of the neural network, the output of $L_i$ is the *preactivation output* of the $i$th layer, and $\sigma \circ L_i$ is the *output* of the $i$th layer. For conciseness, we will write $\nu^{(\ell)}$ to denote the function giving the output of the $\ell$th; i.e.,

$$\nu^{(l)} = \sigma \circ L_l \circ \cdots \circ \sigma \circ L_1.$$

There are many choices for the *activation function $\sigma$*, a popular choice is the rectified linear unit (ReLU) function, $\sigma(t) = \max(0, t)$. Neural networks with ReLU activation will be the main focus of this work, and will usually be referred to simply as *neural networks*.

**Definition A.2.** A set $U \subset \mathbb{R}^n$ of a neural network $f : \mathbb{R}^n \to \mathbb{R}^m$ is a *linear region* if it is a maximal connected region (closure of an open set) on which $f$ is linear.

### A.2  POLYHEDRAL GEOMETRY

Polyhedra are geometric objects described by finitely many inequalities.

**Definition A.3.** A *polyhedron* is a subset of $\mathbb{R}^n$ of the form $P = \{x \in \mathbb{R}^n : Ax \leq b\}$, where $A \in \mathbb{R}^{m \times n}$, $b \in \mathbb{R}^m$, and the inequality is taken element-wise. Such a polyhedron is denoted by $P(A, b)$.

**Lemma A.4.** *Let $P(A, b)$ and $P(A', b')$ be polyhedra. Then*

$$P(A, b) \cap P(A', b') = P\left( \begin{bmatrix} A \\ A' \end{bmatrix}, \begin{bmatrix} b \\ b' \end{bmatrix} \right).$$

*Proof.* Note that for $x \in \mathbb{R}^n$ we have

$$\begin{bmatrix} A \\ A' \end{bmatrix} x = \begin{bmatrix} Ax \\ A'x \end{bmatrix} \leq \begin{bmatrix} b \\ b' \end{bmatrix}$$

if and only if $Ax \leq b$ and $A'x \leq b'$. Therefore, we conclude that

$$P(A, b) \cap P(A', b') = P\left( \begin{bmatrix} A \\ A' \end{bmatrix}, \begin{bmatrix} b \\ b' \end{bmatrix} \right).$$

$\square$

**Example A.5.**

1. $A = \begin{pmatrix} 1 & 1 & 0 \\ 1 & 0 & 1 \end{pmatrix}^\top$ and $b = \begin{pmatrix} 0 & 1 & 1 \end{pmatrix}^\top$ give the the polyhedron in Figure 4a.

2. $A = \begin{pmatrix} 1 & 0 & -1 & 0 \\ 0 & 1 & 0 & -1 \end{pmatrix}^{\top}$ and $b = (1 \quad 1 \quad 1 \quad 1)$ give the polyhedron in Figure 4b.

3. $A = \begin{pmatrix} 1 & 1 & 0 & 1 & 0 & -1 & 0 \\ 1 & 0 & 1 & 0 & 1 & 0 & -1 \end{pmatrix}^{\top}$ and $b = (0 \quad 1 \quad 1 \quad 1 \quad 1 \quad 1 \quad 1)^{\top}$ give the polyhedron in Figure 4c.

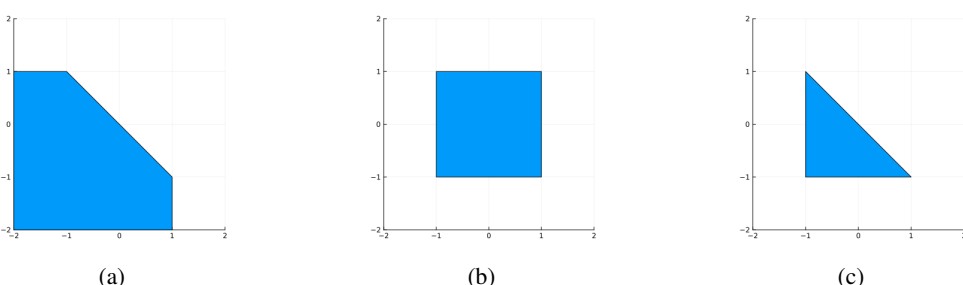

| (a) | (b) | (c) |

Figure 4: Illustrations of the polyhedra constructed in Example A.5.

**Dimension Theory for Polyhedra.**

**Definition A.6.** The *affine hull* of a polyhedron $P \subset \mathbb{R}^n$ is the smallest affine subspace of $\mathbb{R}^n$ that contains $P$, and is denoted by $\mathrm{AffHull}(P)$

**Definition A.7.** The *dimension* of a polyhedron $P$ is the dimension of $\mathrm{AffHull}(P)$.

The description of a polytope as the set of points that satisfy a system comprising finitely many linear inequalities may be needlessly complicated: some inequalities may be redundant and some may be replaced by equalities without changing the set. We now make these notions more precise.

**Definition A.8.** An inequality $\langle \alpha, x \rangle \leq \beta$ in the system $Ax \leq b$ is an *implicit equality* if for any $x$ that satisfies $Ax \leq b$, we have $\langle \alpha, x \rangle = \beta$.

**Notation A.9.** We can partition the system $Ax \leq b$ into two systems: the system of implicit equalities, denoted by $A^{=}x \leq b^{=}$, and the system of remaining inequalities, denoted by $A^{+}x \leq b^{+}$.

The affine hull gives us a convenient way of dealing with the system of implicit equalities.

**Lemma A.10.** *The affine hull of a polyhedron $P = P(A, b)$ admits the following description:*

$$\mathrm{AffHull}(P) = \{x : A^{=}x = b^{=}\}$$

*In particular, the dimension of $P$ is given by*

$$n - \mathrm{rank}(A^{=}).$$

*Proof.* See (Schrijver, 1998, §8.2). $\qquad\square$

Once we know which defining inequalities of a polyhedron $P(A, b)$ are implicit equalities, we can always find a point in the polyhedron that make the remaining inequalities strict.

**Lemma A.11.** *Let $P = P(A, b)$ be a polyhedron. Then there exists a point $\bar{x} \in P$ such that $A^{+}x < b^{+}$.*

*Proof.* See (Schrijver, 1998, §8.1). $\qquad\square$

Algebraic geometry studies geometric properties of solution sets of polynomial systems that can be expressed algebraically, such as their degree, dimension, and irreducible components. *Tropical geometry* is a variant of algebraic geometry where the polynomials are defined in the *tropical semiring*, $\overline{\mathbb{R}} = (\mathbb{R} \cup \{-\infty\}, \oplus, \odot)$ where the addition and multiplication operators are given by $a \oplus b = \max(a, b)$ and $a \odot b = a + b$, respectively. We additionally let $a \oslash b := a - b$.

**Tropical Polynomials.** Using these operations, we can write polynomials as $\bigoplus_m a_m T^m$, where $a_i$ are coefficients, $T \in \overline{\mathbb{R}}$, and where the sum is indexed by a finite subset of $\mathbb{N}$. In our work, we consider the following generalizations of tropical polynomials.

**Definition A.12.** A *tropical Puiseux polynomial* in the indeterminates $T_1, \ldots, T_n$ is a formal expression of the form $\bigoplus_m a_m T^m$ where the index $m$ runs through a finite subset of $\mathbb{Q}_{\geq 0}^n$ and $T^m = T_1^{m_1} \odot \cdots \odot T_n^{m_n}$ with powers taken in the tropical sense.

**Definition A.13.** A *tropical Puiseux rational map* in $T_1, \ldots, T_n$ is a tropical quotient of the form $p \oslash q$ where $p, q$ are tropical Puiseux polynomials.

**Linear Regions of Tropical Polynomials.** Let $f$ be a tropical polynomial in $n$ variables. We write $f = \bigoplus_{j=1}^m a_{\alpha_j} T^{\alpha_j}$ where $\alpha_j = \{\alpha_{j1}, \ldots, \alpha_{jn}\} \in \mathbb{N}^n$ for each $j = 1, \ldots, m$. As a function $\mathbb{R}^n \to \mathbb{R}$, the map $f$ is given by

$$x \mapsto \max_{j \in \{1, \ldots, m\}} \left\{ a_{\alpha_j} + \langle \alpha_j, x \rangle \right\}. \tag{8}$$

Let $i \in \{1, \ldots, m\}$, and consider

$$M_{\alpha_i} = \left\{ x \in \mathbb{R}^n : f(x) = a_{\alpha_i} + \langle \alpha_i, x \rangle \right\},$$

that is $M_{\alpha_i}$ is the subset of points at which the $i$th term is the maximum term in the expression of $f$. Equivalently, we have that

$$
\begin{aligned}
M_{\alpha_i} &= \left\{ x \in \mathbb{R}^n : a_{\alpha_i} + \langle \alpha_i, x \rangle \geq a_{\alpha_j} + \langle \alpha_j, x \rangle \text{ for all } j \in \{1, \ldots, m\} \right\} \\
&= \left\{ x \in \mathbb{R}^n : \langle \alpha_j - \alpha_i, x \rangle \leq a_{\alpha_i} - a_{\alpha_j} \text{ for all } j \in \{1, \ldots, m\} \right\} \\
&= \left\{ x \in \mathbb{R}^n : Ax \leq b \right\},
\end{aligned}
$$

where $A \in \mathbb{R}^{m \times n}$ with $A_{jk} = \alpha_{jk} - \alpha_{ik}$ and $b \in \mathbb{R}^m$ with $b_j = a_{\alpha_i} - a_{\alpha_j}$. That is, the subset $M_\alpha$ is the polyhedron $P(A, b)$.

We then consider the collection of polyhedral $(M_{\alpha_i})_{i \in \{1, \ldots, m\}}$ to determine the linear regions of $f$. That is, we determine the maximally connected sets, which will be some union of the $M_{\alpha_i}$s, such that on these sets $f$ is a linear function.

**Example A.14.**

1. Consider the tropical polynomial $f = 0 \oplus T \oplus T^2$. Then the map $f$ is given by
$$x \mapsto \max \left\{ 0, 1 + x, 1 + 2x \right\},$$
   thus
$$
\begin{cases}
M_0 = P(1, -1) = \{x \leq -1\}, \\
M_1 = P\left([1, -1], \begin{bmatrix} 0 \\ 1 \end{bmatrix}\right) = \{-1 \leq x \leq 0\}, \\
M_2 = P(-1, 0) = \{x \geq 0\}.
\end{cases}
$$
   The linear regions of $f$ are then $\{M_0, M_1, M_2\}$.

2. Consider the tropical polynomial $f = T \oplus 0T^2$. Then the map $f$ is given by
$$x \mapsto \max\{1 + x, 2x\},$$
   thus
$$
\begin{cases}
M_1 = P(1, 1) = \{x \leq 1\} \\
M_2 = P(1, -1) = \{x \geq 1\}.
\end{cases}
$$
   The linear regions of $f$ are then $\{M_1, M_2\}$.

3. Consider the tropical polynomial $f = 1T^0 \oplus T \oplus T^2$. Then the map $f$ is given by

$$x \mapsto \max\{1, 1+x, 1+2x\},$$

thus

$$\begin{cases} M_0 = P(1,0) = \{x \leq 0\}, \\ M_1 = P\left(\begin{bmatrix} 1 \\ -1 \end{bmatrix}, \begin{bmatrix} 0, \\ 0 \end{bmatrix}\right) = \{x = 0\}, \\ M_2 = P(-1,0) = \{x \geq 0\}. \end{cases}$$

The linear regions in this case are $\{M_0, M_2\}$. Note how $M_1$ does not contribute to the map $f$, so the monomial $T$ is redundant. Algorithm 2 would detect this by noting that the dimension of $M_1$ is less than the number of variables of $f$.

4. Consider the tropical polynomial $f = T^2 \oplus T^3 \oplus 2T^4$. Then the map $f$ is given by

$$x \mapsto \max\{1 + 2x,\ 1 + 3x,\ 2 + 4x\},$$

thus

$$\begin{cases} M_2 = P\left(1, -\frac{1}{2}\right) = \left\{x \leq -\frac{1}{2}\right\}, \\ M_3 = \varnothing \\ M_4 = P\left(-1, \frac{1}{2}\right) = \left\{x \geq -\frac{1}{2}\right\}. \end{cases}$$

Hence, the linear regions in this case are $\{M_2, M_4\}$. Note how $M_3$ does not contribute to the map $f$. Hence, the monomial $T^3$ is redundant. One of the results of Section D, Lemma 4.1, shows that this is related to the fact that the dimension of $M_3$ is less than the number of variables of $f$.

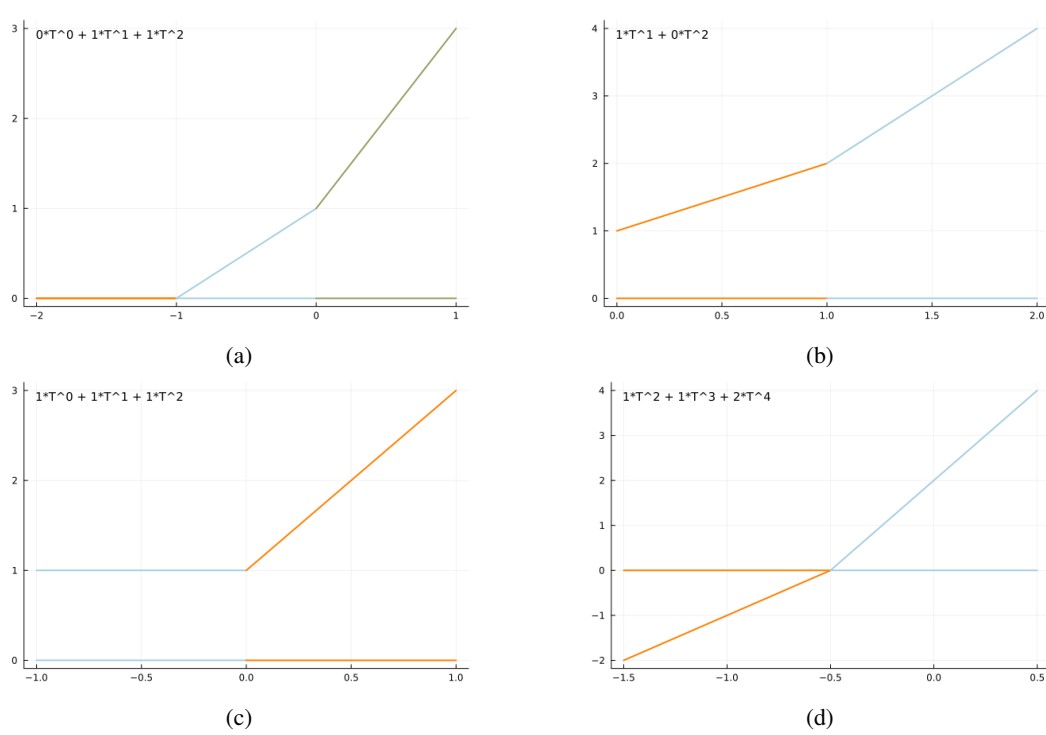

Figure 5: Illustrations of the linear regions, depicted as horizontal lines on the horizontal axis, and the corresponding linear maps of the tropical polynomials of Example A.14.

For a tropical polynomial $f$, a point that lies in two linear regions is a point at which the maximum identified in equation 8 is attained at two different terms in the expression of $f$. In standard tropical geometry terminology, these points are precisely those found on the tropical hypersurface cut of by $f$.

**Tropical Expressions for Neural Networks.** The first explicit connection between tropical geometry and neural networks was established in Zhang et al. (2018); we adopt a similar notation.

One of the key observations for intersecting tropical geometry and deep learning is that, up to rescaling of rational weights to obtain integers, neural networks can be written as tropical rational functions (Zhang et al., 2018, Theorem 5.2). From a more computational perspective, it is usually preferable to avoid such rescaling and simply work with the original weights. The proof of Theorem 5.2 in Zhang et al. (2018) can be directly adapted to show that any neural network can be written as the function associated to a tropical Puiseux rational map. In their language, this corresponds to saying that any neural network is a *tropical rational signomial* with nonnegative rational exponents.

*Proof.* Let $f : \mathbb{R}^n \to \mathbb{R}^m$ be a neural network with architecture $[n_0, \ldots, n_d]$. Let $A_1 = [a_{ij}] \in \mathbb{Z}^{n_0 \times n_1}$ and $b \in \mathbb{R}^{n_1}$ be the weight and bias vectors respectively for the first layer of the network. Let $(A_1)_+ = \left[a_{ij}^+\right]$ where $a_{ij}^+ := \max\{a_{ij}, 0\}$, and let $(A_1)_- = \left[a_{ij}^-\right]$ where $a_{ij}^- := \max\{-a_{ij}, 0\}$. So that $(A_1)_+, (A_1)_- \in \mathbb{N}^{m \times n}$ with $A_1 = (A_1)_+ - (A_1)_-$. Observe,
$$\sigma(A_1 x + b) = \max(0, A_1 x + b) = \sigma((A_1)_+ x + b, (A_1)_- x) - A_- x.$$
That is, every coordinate of the output of the first layer of the network can be written as the difference of tropical polynomials. Assume this is also true for the $l^{\text{th}}$ layer, where $l < d$, so we can write
$$\nu^{(l)}(x) = F^{(l)}(x) \oslash G^{(l)}(x),$$
where $F^{(l)}$ and $G^{(l)}(x)$ are tropical polynomials. Then
$$
\begin{aligned}
L_{l+1} \circ \nu^{(l)}(x) &= \left((A_{l+1})_+ - (A_{l+1})_-\right)\left(F^{(l)}(x) - G^{(l)}(x)\right) + b_l \\
&= \left((A_{l+1})_+ F^{(l)}(x) + (A_{l+1})_- G^{(l)}(x) + b_{l+1}\right) \\
&\quad - \left((A_{l+1})_- G^{(l)}(x) + (A_{l+1})_- F^{(l)}(x)\right) \\
&= H^{(l+1)}(x) - G^{(l+1)}(x),
\end{aligned}
$$
where $H^{(l+1)}(x)$ is a tropical polynomial. Therefore, the output of the $(\ell + 1)$st can be written as
$$
\begin{aligned}
\nu^{(\ell+1)}(x) &= \max\left\{0, \, L_{\ell+1} \circ \nu^{(l)}\right\} \\
&= \max\left\{0, \, H^{(\ell+1)}(x) - G^{(\ell+1)}(x)\right\} \\
&= \max\left\{H^{(\ell+1)}(x), \, G^{(\ell+1)}(x)\right\} - G^{(\ell+1)}(x) \\
&= F^{(\ell+1)}(x) - G^{(\ell+1)}(x),
\end{aligned}
$$
where, as before, $F^{(\ell+1)}$ and $G^{(\ell+1)}$. Hence, through inductive arguments, we deduce that the neural network $f$ can be written as a difference of tropical polynomials, that is $f$ can be written as a tropical rational map. $\qquad\square$

The proof that neural networks can be written as tropical rational maps, provides a recursive construction that we utilise to computationally obtain a tropical representation of a neural network.

### A.4 PERMUTATION INVARIANT NEURAL NETWORKS

Informally, a permutation invariant neural network is a neural network whose output is unchanged upon re-ordering its inputs.

**Definition A.15.** A *permutation matrix* $P \in \mathbb{R}^{n \times n}$ is a matrix of zeros with exactly one entry equal to one in each row and column.

**Example A.16.** $P_1 = \begin{pmatrix} 0 & 1 & 0 \\ 1 & 0 & 0 \\ 0 & 0 & 1 \end{pmatrix}$ is a permutation matrix, whereas $P_2 = \begin{pmatrix} 1 & 1 & 0 \\ 0 & 0 & 1 \\ 1 & 0 & 0 \end{pmatrix}$ is not since the first row contain two ones. Note that
$$P_1 \begin{pmatrix} x_1 \\ x_2 \\ x_3 \end{pmatrix} = \begin{pmatrix} x_2 \\ x_1 \\ x_3 \end{pmatrix}.$$

That is, left multiplication by $P_1$ has the effect of permuting the entries of the vector.

**Definition A.17.**

1. A function $f : \mathbb{R}^n \to \mathbb{R}^m$ is *permutation invariant* if $f(Px)$ for every $n \times n$ permutation matrix $P$.

2. A function $f : \mathbb{R}^n \to \mathbb{R}^n$ is *permutation equivariant* if $f(Px) = Pf(x)$ for every $n \times n$ permutation matrix $P$.

**Lemma A.18** (Zaheer et al. (2017))**.** *An $m \times m$ matrix $W$ acting as a linear operator of the form $W = \lambda I_{m \times m} + \gamma(\mathbf{1}^\top \mathbf{1})$, where $\lambda, \gamma \in \mathbb{R}$ and $I_{m \times m}$ is the $m \times m$ identity matrix, is permutation equivariant.*

Now consider the neural network $f : \mathbb{R}^n \to \mathbb{R}$ given by

$$f(x) = \sum_{i=1}^{n} \sigma(Wx), \tag{9}$$

where $\sigma$ is the ReLU activation function and $W$ is as in Lemma A.18. Then $f$ is permutation invariant since

$$f(Px) = \sum_{i=1}^{n} \sigma(WPx) \stackrel{\text{Lem}A.18}{=} \sum_{i=1}^{n} P\sigma(Wx) = \sum_{i=1}^{n} \sigma(Wx) = f(x),$$

where we have used the fact that the ReLU activation is applied element-wise, and summation is a permutation invariant operation.

**Lemma A.19.** *The set*

$$\Delta = \{(x_1, \ldots, x_n) : x_1 \geq x_2 \geq \cdots \geq x_n\}$$

*is a fundamental domain of the action of $S_n$ on $\mathbb{R}^n$ that permutes coordinates.*

*Proof.* It suffices to show that conditions (i) and (ii) of Definition 3.2 are satisfied.
(i): Let $x = (x_1, \ldots, x_n) \in \mathbb{R}^n$. Let $g : \{1, \ldots, n\} \to \{1, \ldots, n\}$ be a bijective function such that

$$x_{g(1)} \geq x_{g(2)} \geq \cdots \geq x_{g(n)}.$$

Then $g \in S_n$, and $g \cdot x = (x_{g(1)}, \ldots, x_{g(n)}) =: \hat{x} \in \Delta$. Therefore,

$$x = g^{-1} \cdot \hat{x} \in \bigcup_{h \in S_n} h \cdot \Delta.$$

Thus since clearly $\bigcup_{g \in S_n} g \cdot \Delta \subset \mathbb{R}^n$, we deduce that $\mathbb{R}^n = \bigcup_{g \in G} g \cdot \Delta$.
(ii): Let $g, \in S_n$ be distinct. Suppose for contradiction that $x \in (g \cdot \text{int}(\Delta) \cap h \cdot \text{int}(\Delta))$. Then $x = g \cdot y$ and $x = h \cdot z$, for some $y, z \in \text{int}(\Delta)$. In particular, $y$ and $z$ are such that

$$y_1 > y_2 > \cdots > y_n \text{ and } z_1 > z_2 > \cdots > z_n.$$

We know that for each $j \in \{1, \ldots, m\}$ we have $y_j = z_k$ for some $k \in \{1, \ldots, m\}$. Suppose $y_j = z_k$ then

$$y_1 > \ldots y_{j-1} > y_j = z_k > z_{k+1} > \cdots > z_n.$$

This means the $y_1, \ldots, y_{j-1}$ can only pair with the $z_1, \ldots, z_{k-1}$, and thus $j = k$. Therefore, $y = z$ which implies that $g = h$ which is a contradiction. We conclude then that $g \cdot \text{int}(\Delta) \cap h \cdot \text{int}(\Delta) = \varnothing$. $\qquad\square$

# B SYMBOLIC ALGORITHMS

---

**Algorithm 1** Linear regions of tropical Puiseux rational functions

---

**Require:** Tropical Puiseux polynomials $p, q$ in $n$ variables.
1: Compute the linear regions $U_1, \ldots, U_\ell$ of $p$, and set $L_i = L(p, U_i)$.
2: Compute the linear regions $V_1, \ldots, V_m$ of $q$, and set $S_j = L(q, V_j)$.
3: Compute the pairs $(i, j)$ such that $U_i \cap V_j$ has dimension $n$
4: **for** $(i, j)$ such that $U_i \cap V_j$ has dimension $n$ **do**
5:     Compute the linear map $T_{ij} = L_i - S_j$

6: Set $S$ to be the set of all $T_{ij}$
7: **for** $T \in S$ **do**
8:     Compute the set $I(T)$ indices $(i, j)$ such that $T = T_{ij}$.
9:     Compute the set $C(T)$ of connected components of

$$\bigcup_{(i,j)\in I(T)} U_i \cap V_j$$

    **return** $\bigcup_{T \in S} C(T)$.

---

---

**Algorithm 2** Pruning tropical expressions.

---

**Require:** Tropical Puiseux polynomial $g$ in $n$ variables.
1: **for** for each monomial $a_i T^i$ **do**
2:     Compute the corresponding polytope $P_i$.
3:     **if** $P_i$ has dimension less than $n$ **then**
4:         Discard the $i$th monomial
    **return** g

---

# C NUMERICAL ESTIMATION OF LINEAR REGIONS

## C.1 OVERVIEW

The method of numerical estimation we use is inspired by the recent work of Goujon et al. (2024). Specifically, to numerically estimate the number of linear regions of a neural network, we exploit the fact that the linear regions of a neural network correspond to regions where the gradient is constant. We evaluate the gradient on a sample of points in some bounded region $X$ and identify the number of unique gradients we obtain. Care needs to be taken at this step, since it may be the case that the same linear function operates on disconnected regions. In our symbolic approach, this would correspond to distinct linear regions. To try and account for this in our numerical approach, for points with the same Jacobian, we sample the model at their midpoint and compare this to the midpoint provided by the linear map. If these values differ, then the regions are disconnected. However, if the values do not differ, we still cannot be certain whether the region is connected or disconnected. Therefore, our numerical approach is likely to *underestimate* the number of linear regions in this instance. We summarize this technique with Algorithm 3.

There are a few sources of errors that arise in our method of numerical approximation that are important to note:

1. We cannot be sure if our search radius captures all of the linear regions.

2. It may be the case that disconnected regions are acted on by the same linear map. The symbolic approach would count these regions as distinct. In our numerical approach, we try to resolve this by additionally sampling at the midpoint of points with the same Jacobian, however, this does not guarantee that we identify disconnected regions.

3. From exploratory experiments, we observe that some linear regions are very small. Therefore, a highly refined grid would be required to identify them. As the dimension of the

---

**Algorithm 3** Numerical estimation of neural network linear regions

---

**Require:** A linear activation neural network $f$ with scalar output, a bounded subset of the input
  domain $X$, $N$ number of points to sample.
 1: Sample $N$ points from $X$
 2: Compute the Jacobian matrices of the network at each point.
 3: Round the Jacobians matrices to 10 decimal places to avoid floating point errors.
 4: Count the Jacobians that appear uniquely.
 5: **for** Duplicate Jacobians **do**
 6:  Obtain the corresponding sample midpoint.
 7:  Obtain the midpoint of the model output at the sample points.
 8:  **if** Model at the sample midpoint is equal to the midpoint of the output midpoints **then**
 9:    Count the duplicate as a single linear region.
 10:  **else**
 11:    Count the duplicate as separate linear regions.
   **return** The number of linear regions.

---

    input increases, we require exponentially more points to maintain a certain density level,
    which quickly becomes infeasible.

## C.2 COMPARISON WITH THE SYMBOLIC APPROACH

Our contributions provide a symbolic approach for computing the linear regions of a given neural
network. Specifically, we compute the tropical expression for a neural network and then use Algo-
rithm 1 to compute its linear regions. The advantage of this approach is that we obtain an exact
characterization of the linear regions of a neural network. However, as expected and as we will
show, this approach is more computationally expensive and thus takes more time than numerical
approaches such as Algorithm 3. Therefore, in practical situations, numerical approaches may still
be preferred. However, our symbolic method can be used to assess the precision of these numerical
approaches by comparing them to the ground truth.

Here we use both Algorithm 1 and Algorithm 3 to obtain the number of linear regions of neural
networks of different architectures. We implement Algorithm 3 with $X$ as a cube of radius $R$ for
multiple values of $R$ and using various sample sizes, due to the lack of an efficient method to set
these parameters optimally. For each algorithm and configuration, we sample 25 neural networks.
The results are presented in Table 1. For the tables containing the results of the numerical approach,
$N$ denotes the number of points and $R$ denotes the search radius.

We find here that the numerical approximations are on par with the symbolic computations but have
the advantage of running faster. However, this precision is not guaranteed and we see it deteriorates
for neural networks with larger architectures, probably as a consequence of some of the issues out-
lined above. One particular issue is too small a search radius, which cannot capture all of the linear
regions. Indeed, for 4-layered networks, increasing the search radius improves the approximation.

## C.3 IMPLEMENTATION ON INVARIANT NETWORKS

We can apply this method of numerical sampling to assess the optimisation provided by our introduc-
tion of a fundamental domain for invariant neural networks. In Figure 6, we present a comparison
between the estimate of the number of linear regions of a permutation invariant when we utilize the
fundamental domain and when we do not. We first initialize a permutation invariant network with
$n$ input dimensions. Then we apply Algorithm 3 with $X = [-20, 20]^n$ and $N = 10^n$ to obtain an
estimate that does not account for the symmetries of the network. To account for the symmetries, we
instead apply Algorithm 3 with $X = [-20, 20]^n \cap \Delta$, $N = \frac{10^n}{n!}$ and account for the multiplicities
using Lemma 3.4. We record the ratio of these estimates as well as the ratios of their execution times.
Figure 6 shows the average of these values across 10 iterations of this procedure.

We observe that the fundamental domain estimate performs well for low dimensional inputs and
provides significant improvements in execution time. Despite the divergence as the input dimension
increases, this estimate is still useful because we are often more concerned with obtaining an upper

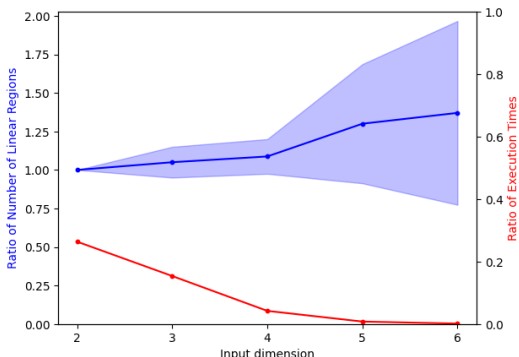

Figure 6: Ratio estimates for different input sizes with standard deviation error bars.

bound on the complexity of a neural network rather than an exact figure and the fundamental domain estimate does not undercount the number of linear regions.

## D  MONOMIALS AS A MEASURE OF COMPLEXITY

Various measures of neural network complexity have been proposed and studied, such as counting the linear regions of ReLU neural networks (Montúfar et al., 2014); measuring the effect of forward propagation on the length of 1D curves (Raghu et al., 2016); and evaluating the sum of the Betti numbers of decision regions (Bianchini et al., 2014). Our work focuses on the number of linear regions of the input domain partitioned by the neural network. We provide tools to evaluate this measure exactly by capitalizing on the representation of neural networks as tropical Puiseux rational functions. As a consequence of this approach, we obtain another measure of neural network complexity, namely the number of monomials in the tropical expression of the neural network which quantifies its *algebraic* complexity.

**Definition D.1.** Let $f$ be a neural network, and $g \oslash h$ a tropical representation of $f$, i.e., a tropical Puiseux rational function whose underlying real-valued function equals $f$. If $g$ has $m$ irredundant monomials and $h$ has $n$ irredundant monomials then we define the *monomial complexity* of the representation $g \oslash h$ to be the pair $(m, n)$.

Intuitively, this captures how many linear terms needed to express the neural network. Notice that this measure is closely related to, but not identical to, the number of linear regions of a neural network.

**Example D.2.**

1. If $g$ is a tropical polynomial then the number of irredundant monomials of $g$ is equal to the number of linear regions of $g$.

2. If $f$ is a neural network with a tropical representation of monomial complexity $(m, n)$ then $f$ has at most $mn$ linear regions.

We emphasize that the number of linear regions and the number of monomials of a tropical expression – which we just take to be the sum of the monomials present in the numerator and denominator – are linked, but distinct. To provide intuition for how these two quantities are connected, Figure 7 shows the evolution of the number of linear regions as we vary the number of monomials of randomly generated Puiseux rational functions in 3 and 4 variables.

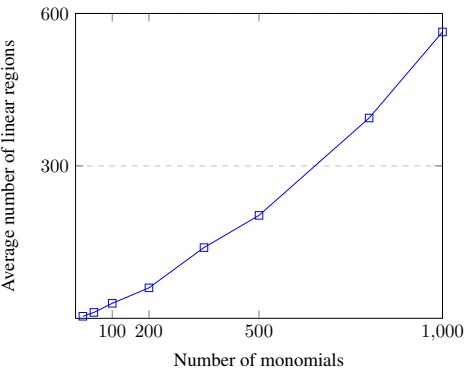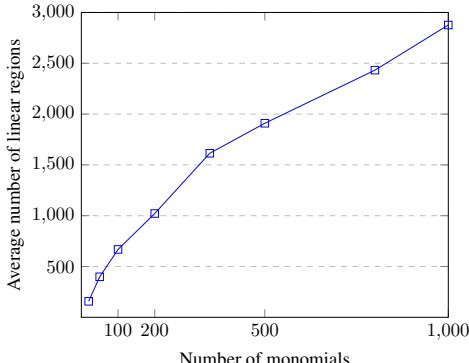

Figure 7: **Left:** Linear regions of a Puiseux rational function in 3 variables. **Right:** Linear regions of a Puiseux rational function in 4 variables.

# E  COMPUTING AND ESTIMATING HOFFMAN CONSTANTS

**The PVZ Algorithm.** In Peña et al. (2018), the authors proposed a combinatorial algorithm to compute the precise value of the Hoffman constant for a matrix $A \in \mathbb{R}^{m \times n}$, which we refer to as the *Peña–Vera–Zuluaga (PVZ) algorithm* and sketch its main steps here.

**Definition E.1.** A set-valued map $\Phi : \mathbb{R}^n \to \mathbb{R}^m$ assigns a set $\Phi(x) \subseteq \mathbb{R}^m$. The map is surjective if $\Phi(\mathbb{R}^n) = \cup_x \Phi(x) = \mathbb{R}^m$. Let $A \in \mathbb{R}^{m \times n}$. For any $J \subseteq \{1, 2, \ldots, m\}$, let $A_J$ be the submatrix of $A$ consisting of rows with indices in $J$. The set $J$ is called *$A$-surjective* if the set-valued map $\Phi(x) = A_J x + \{y \in \mathbb{R}^J : y \geq 0\}$ is surjective.

Notice that $A$-surjectivity is a generalization of linear independence of row vectors. We illustrate this observation in the following two examples.

**Example E.2.**

1. If $J$ is such that $A_J$ is full-rank, then $J$ is $A$-surjective, since for any $y \in \mathbb{R}^J$, there exists $x \in \mathbb{R}^n$ such that $y = A_J x$.

2. Let $A = \mathbf{1}_{m \times n}$ be the $m \times n$ matrix whose entries are 1's. For any subset $J$ of $\{1, \ldots, m\}$ and for any $y \in \mathbb{R}^J$, let $x \in \mathbb{R}^n$ such that $\sum_i x_i \leq \min\{y_j, j \in J\}$. Then $y - A_J x \geq 0$. Thus any $J$ is $A$-surjective.

The PVZ algorithm is based on the following characterization of the Hoffman constant.

**Proposition E.3.** *(Peña et al., 2018, Proposition 2) Let $A \in \mathbb{R}^{m \times n}$. Equip $\mathbb{R}^m$ and $\mathbb{R}^n$ with norm $\| \cdot \|$ and denote its dual norm by $\| \cdot \|^*$. Let $\mathcal{S}(A)$ be the set of all $A$-surjective sets. Then*

$$H(A) = \max_{J \in \mathcal{S}(A)} H_J(A) \tag{10}$$

*where*

$$H_J(A) = \max_{y \in \mathbb{R}^m, \|y\| \leq 1} \min_{\substack{x \in \mathbb{R}^n \\ A_J x \leq y_J}} \|x\| = \frac{1}{\min_{v \in \mathbb{R}^J_+, \|v\|^* = 1} \|A_J^\top v\|^*}. \tag{11}$$

This characterization is particularly useful when $\mathbb{R}^m$ and $\mathbb{R}^n$ are equipped with the $\infty$-norm, since the computation of equation 11 reduces to a linear programming (LP) problem. The key problem is how to maximize over all $A$-surjective sets. To do this, the PVZ algorithm maintains three collections of sets $\mathcal{F}$, $\mathcal{I}$, and $\mathcal{J}$ where during every iteration: (i) $\mathcal{F}$ contains $J$ such that $J$ is $A$-surjective; (ii) $\mathcal{I}$ contains $J$ such that $J$ is not $A$-surjective; and (iii) $\mathcal{J}$ contains candidates $J$ whose $A$-surjectivity will be tested.

To detect whether a candidate $J \in \mathcal{J}$ is surjective, the PVZ algorithm requires solving

$$\min \left\| A_J^\top v \right\|_1, \ s.t. \ v \in \mathbb{R}_+^J, \|v\|_1 = 1. \tag{12}$$

If the optimal value is positive, then $J$ is $A$-surjective, and $J$ is assigned to $\mathcal{F}$ and all subsets of $J$ are removed from $\mathcal{J}$. Otherwise, the optimal value is 0 and there is $v \in \mathbb{R}_+^J$ such that $A_J^\top v = 0$. Let $I(v) = \{i \in J : v_i > 0\}$ and assign $I(v)$ to $\mathcal{I}$. Let $\hat{J} \in \mathcal{J}$ be any set containing $I(v)$. Replace all such $\hat{J}$ by sets $\hat{J} \backslash \{i\}, i \in I(v)$ which are not contained in any sets in $\mathcal{F}$.

**Lower and Upper Bounds.** A limitation of the PVZ algorithm is that during each loop, every set in $\mathcal{J}$ needs to be tested, and each test requires solving an LP problem. Although solving one LP problem in practice is fast, a complete while loop calls the LP solver many times.

Here, we propose algorithms to estimate lower and upper bounds on the Hoffman constant. An intuitive way to estimate the lower bound is to sample a number of random subsets from $\{1, \ldots, m\}$ and test for $A$-surjectivity. This method bypasses optimizing combinatorially over $\mathcal{S}(A)$ of $A$-surjective sets and gives a lower bound of Hoffman constant by Proposition E.3.

To get an upper bound on the Hoffman constant, we use the following result from Güler et al. (1995).

**Theorem E.4.** *(Güler et al., 1995, Theorem 4.2) Let $A \in \mathbb{R}^{m \times n}$. Let $\mathcal{D}(A)$ be a set of subsets of $J \subseteq \{1, \ldots, m\}$ such that $A_J$ is full rank. Let $\mathcal{D}^*(A)$ be the set of maximal elements in $\mathcal{D}(A)$. Then the Hoffman constant measured under 2-norm is bounded by*

$$H(A) \leq \max_{J \in \mathcal{D}^*(A)} \frac{1}{\hat{\rho}(A_J)} \tag{13}$$

*where $\hat{\rho}(A)$ is the smallest singular value of $A$.*

Using the fact that $\| \cdot \|_1 \geq \| \cdot \|_2$, and the characterization from equation 11, we see that the upper bound also holds when $\mathbb{R}^m$ and $\mathbb{R}^n$ are equipped with the $\infty$-norm. We use equation 13 as an approximation to the solution of the LP problem in equation 12. Although this replaces solving an LP problem with finding singular values, which is a lot more efficient in practice, it still requires a combinatorial search.

---

**Algorithm 4** Exact computation of the Hoffman constant

---

**Require:** $A$: an $m \times n$ matrix
 1: Initialize $H = 0$.
 2: **for** subset $J$ of all subsets of $\{1, \ldots, m\}$ **do**
 3:    Solve equation 12. Let $t$ be the optimal value;
 4:    **if** $t > 0$ **then**
 5:       $J$ is surjective. Update $H = \max\left\{H, \frac{1}{t}\right\}$;
    **return** Hoffman constant $H$.

---

**Algorithm 5** Lower bound for Hoffman constant

---

**Require:** $A$: an $m \times n$ matrix, $B$ number of iterations
 1: Initialize $H_L = 0$.
 2: **for** $i \in \{1, \ldots, B\}$ **do**
 3:    Sample a random integer $K$.
 4:    Sample a random subset $J$ for $\{1, \ldots, m\}$ of size $K$.
 5:    Solve equation 12. Let $t$ be the optimal value;
 6:    **if** $t > 0$ **then**
 7:       $J$ is surjective. Update $H_L = \max\left\{H_L, \frac{1}{t}\right\}$;
    **return** Lower bound for Hoffman constant $H_L$.

---

**Numerical Verification.** We verify our approaches on synthetic data. More specifically, we generate Puiseux rational maps by randomly generating two tropical Puiseux polynomials $p$ and $q$, with $m_p$ and $m_q$ monomials respectively. We do so by constructing an $m_p \times n$ matrix $A_p$ and an $m_q \times n$

---

| | |
|---|---|
| 1242 | **Algorithm 6** Upper bound for Hoffman constant |
| 1243 | |

**Require:** $A$: an $m \times n$ matrix
1: Initialize $H_U = 0$.
2: **for** subset $J$ of all subsets of $\{1, \ldots, m\}$ **do**
3:     Compute the minimal singular value of $\hat{\rho}(A_J)$
4:     **if** $\hat{\rho}(A_J) > 0$ **then**
5:         Update $H_U = \max\left\{H_U, \frac{1}{t}\right\}$;
    **return** Upper bound for Hoffman constant $H_U$.

---

matrix $A_q$ by uniformly sampling entries from $[0,1]$. We then form the matrix of equation 4 and compute the exact Hoffman constant along with approximations of its lower and upper bound by our proposed Algorithms 5 and 6. However, upon careful investigations of the public code provided by Peña et al. (2018), we find the output numerical values are unstable. To complete our experiments, we then tested examples without using the public code, and instead implemented a brute force computation by computing equation 12 over all submatrices. The brute force approach is given by Algorithm 4.

For the combination of different values $m_p$, $m_q$, $n$ and $B$, we repeat all computations 8 times. The true Hoffman constants, lower bounds, upper bounds, and the computation time can be found in Tables 3. Although we did not use PVZ algorithm to compute the exact values, for the sake of completeness, we also record the computation time and the number of calls to solve the LP problem within the loop of the PVZ algorithm. The number of iterations of the PVZ algorithm with the average time to solve the LP problems during each stage can be found in Tables 2a,2b,2c. From the tables, we see that computing the true Hoffman constants requires solving over 1000 LP problems, which is computationally expensive. Although the lower and upper bounds can be loose, the computational times are much faster, which illustrates their practicality in real data applications.

# F   PROOFS

## F.1   PROOF OF PROPOSITION 2.4

*Proof.* The polyhedra defined by

$$\left(\begin{bmatrix} A \\ A' \end{bmatrix} - \mathbf{1}\begin{pmatrix} a_i \\ a'_j \end{pmatrix}\right) x \leq \begin{bmatrix} b_i\mathbf{1} - b \\ b'_j\mathbf{1} - b' \end{bmatrix}$$

form a convex refinement of linear regions of $f$. Let

$$\text{res}_{i,j}(x) := \left(\begin{bmatrix} A \\ A' \end{bmatrix} - \mathbf{1}\begin{bmatrix} a_i \\ a'_j \end{bmatrix}\right) x - \begin{bmatrix} b_i\mathbf{1} - b \\ b_j\mathbf{1} - b' \end{bmatrix}$$

denote the residual of $x$ to the polyhedron. We have

$$R_f(x) \leq H(p \oslash q) \max\left\{\|\text{res}_{i,j}(x)_+\|_\infty : 1 \leq i \leq m_p\,; 1 \leq j \leq m_q\right\}.$$

Note that

$$\|\text{res}_{i,j}(x)_+\|_\infty = \left\|\left(\begin{bmatrix} Ax + b - \mathbf{1}(a_ix + b_i) \\ A'x + b' - \mathbf{1}(a'_jx + b'_j) \end{bmatrix}\right)_+\right\|_\infty$$

$$= \max_{k,\ell}\left\{(Ax+b)_k - (a_ix+b_i),\, (A'x+b')_\ell - (a'_jx+b'_j),\, 0\right\}$$

$$= \max_{i,j}\left\{p(x) - (a_ix+b_i),\, q(x) - (a'_jx+b'_j),\, 0\right\}$$

Therefore,

$$\max_{i,j}\|\text{res}_{i,j}(x)\|_\infty = \max_{i,j}\left\{p(x) - (a_ix+b_i),\, q(x) - (a'_jx+b'_j),\, 0\right\}$$

$$= \max\left\{p(x) - \min_i\{a_ix+b_i\},\, q(x) - \min_j\{a'_jx+b'_j\},\, 0\right\}$$

$$= \max\left\{p(x) - \check{p}(x),\, q(x) - \check{q}(x)\right\}$$

which proves equation 6. $\qquad\qquad\square$

### F.2   PROOF OF LEMMA 2.3

*Proof.* From the definition of minimal effective radius, we have

$$R_f(x) = \min\left\{r : B(x, r) \cap U_i \neq \varnothing, U_i \in \mathcal{U}\right\}$$
$$= \min\{r : d(x, U_i) \leq r, U_i \in \mathcal{U}\}$$
$$= \max\{d(x, U_i) : U_i \in \mathcal{U}\}.$$

For each linear region $U_i$ characterized by $\widetilde{A}_{U_i} x \leq \widetilde{b}_{U_i}$, by equation 1 we have

$$d(x, U_i) \leq H(\widetilde{A}_{U_i}) \| (\widetilde{A}_{U_i} x - \widetilde{b}_{U_i})_+ \|.$$

Passing to the maximum we have

$$R_f(x) = \max_{U_i \in \mathcal{U}} d(x, U_i)$$
$$\leq \max_{U_i \in \mathcal{U}} H(\widetilde{A}_{U_i}) \max_{U_i \in \mathcal{U}} \left\| (\widetilde{A}_{U_i} x - \widetilde{b}_{U_i})_+ \right\|$$
$$= H(f) \max_{U_i \in \mathcal{U}} \left\| (\widetilde{A}_{U_i} x - \widetilde{b}_{U_i})_+ \right\|.$$

$\square$

### F.3   PROOF OF THEOREM 3.3

*Proof.* For any linear region $A$, we denote the orbit of $A$ by $[A]$. The action of $G$ partitions $\mathcal{U}$ into a set of orbits $[\mathcal{U}]$, and thus

$$|\mathcal{U}| = \sum_{[A] \in [\mathcal{U}]} |[A]|.$$

From property (i) in the definition of a fundamental domain, we have

$$\bigcup_{A \in \mathcal{U}} A = \bigcup_{\sigma \in G} \sigma \cdot \Delta,$$

which implies the following estimate:

$$|\mathcal{U}| \geq \sum_{A \in \mathcal{U}_c} |[A]| = |G| |\mathcal{U}_c|.$$

For any $A \in \mathcal{U}$, the orbit stabilizer theorem states that $|[A]||G_A| = |G|$. Thus we have

$$|\mathcal{U}| \leq \sum_{A \in \mathcal{U}_e} |[A]| \leq |G| |\mathcal{U}_c| + \sum_{A \in \mathcal{U}_e \setminus \mathcal{U}_c} \frac{|G|}{|G_A|}.$$

$\square$

**Theorem F.1** (Correctness of Algorithm 1). *Algorithm 1 computes the exact number of linear regions of a Puiseux rational function $f = p \oslash q$.*

*Proof.* Let $U_1, \ldots, U_\ell$ be the linear regions of $p$ and $L_i = L(p, U_i)$. Similarly let $V_1, \ldots, V_m$ be the linear regions of $q$, and set $S_j = L(q, V_j)$. We take $T_{ij} = L_i - S_j$ and set

$$S = \{T_{ij} \mid U_i \cap V_j \text{ has dimension } n\}.$$

For $T \in S$, let $I(T)$ be the set of pairs $(i, j)$ such that $T = T_{ij}$, and $C(T)$ the connected components of

$$\bigcup_{(i,j) \in I(T)} U_i \cap V_j.$$

We need to check that the set of linear regions of $f$ is precisely the union

$$\mathcal{U} = \bigcup_{T \in S} C(T).$$

It suffices to check that:

(i) The elements of this set cover $\mathbb{R}^n$;

(ii) $f$ is linear on each region in $\mathcal{U}$; and

(iii) Each element $D$ in $\mathcal{U}$ is maximal in the sense that there is no (connected) region $E$ containing $D$ as a strict subset such that $f$ is linear on $E$.

(i) follows from the fact that the sets $\{U_i \cap V_j \,|\, \dim U_i \cap V_j = n\}$ cover $\mathbb{R}^n$; (ii) holds because by definition, any element of $\mathcal{U}$ is a subset of

$$\bigcup_{(i,j) \in I(T)} U_i \cap V_j$$

for some $T$ and the set of indices $I(T)$ was constructed precisely so that $f$ can be represented by the linear map $T$ on this union. For (iii) it suffices to notice that

$$\bigcup_{(i,j) \in I(T)} U_i \cap V_j = \{x \in \mathbb{R}^n \,|\, \text{There exists an open } N \text{ such that } x \in \overline{N}, \text{ and } f|_N = T|_N\}$$

so connected components of this union are maximal connected regions of $\mathbb{R}^n$ on which $f$ is linear.

$\square$

## F.4 PROOF OF LEMMA 3.4

*Proof.* Note that since $f$ is permutation invariant, the Jacobian at $Px$, for a permutation matrix $P$, is equal to $J$. If $J$ has distinct elements, then the region is contained within the interior of the fundamental region, and thus by property (ii) of Definition 3.2 we obtain $n!$ factorial distinct regions with Jacobian $J$. On the other hand, if $J$ has an entry repeated $m$ times, then the region is symmetric under $m!$ permutations of $S_n$. Thus, there exists at most $\frac{n!}{m!}$ regions with Jacobian $J$, since some transformed regions may be connected and thus not be distinguished as separate linear regions. Generalizing this argument, it follows that a given Jacobian $J$ corresponds to at most

$$\text{mult}(J) = \frac{n!}{\prod_{c \in C(J)} c!}$$

linear regions.

$\square$

## F.5 PROOF OF LEMMA 4.1

*Proof.* Let $g = \bigoplus_i a_i T^{\alpha_i}$ be a tropical Puiseux polynomial, and let $P_k$ be the polytope associated to the $k$th monomial of the expression of $g$. Recall that $P_k$ is defined by the system of linear inequalities $(*)$

$$\langle \alpha_j, x \rangle + a_j \leq \langle \alpha_k, x \rangle + a_k \text{ for all } j \neq k.$$

Assume $\dim P_k < n$. Then, the system $(*)$ must contain at least one implicit equality

$$\langle \alpha_j, x \rangle + a_j \leq \langle \alpha_k, x \rangle + a_k$$

such that $\alpha_j \neq \alpha_k$. In particular, the maximum in $g(x) = \max_i \langle \alpha_i, x \rangle + a_i$ is attained at the $j$th term whenever it is attained at the $k$th term, and we can remove the $k$th term from the expression without modifying the corresponding function.

Conversely, let us assume that we can remove the $k$th monomial from the expression of $g$ without changing the corresponding function. Suppose for a contradiction that $P_k$ has dimension $n$. This implies that the system has no implicit equalities, and thus by Lemma A.11 we can find a point $\bar{x} \in P_k$ such that all the inequalities in $(*)$ are strict. This contradicts our assumption on the redundancy of the $k$th monomial.

$\square$

# G FURTHER EXPERIMENTAL DETAILS

Experiments were run on an i7-1165G7 CPU with 16GB of RAM. Table 4 lists the time taken by each experiment. Given that our experiments do not include training on large datasets, the experiments are not particularly expensive from the perspective of memory usage, and all the code can be run on a laptop. The detail provided in the paper corresponds roughly to the amount of computational resources that were used for this work, omitting trial and testing runs.

The code used to run the experiments, including the Julia library forming a part of our symbolic contribution, can be found in the following anonymized repository:
https://anonymous.4open.science/r/tropical-expressivity/README.md

# H COMPUTATIONAL DEMONSTRATION

We randomly initialize a ReLU neural network of architecture $[2, 6, 1]$, and deduce various properties using our tools. The native tropical representation of the network is

$$
\left(\frac{8303211062024807}{18014398509481984}x_1^{\frac{5315625921210391264531289425775}{2028240960365167042394725128601}}x_2^{\frac{4280731817019129183430556 4285899}{8112963841460668169578900514406}}\right.
$$

$$
\oplus \frac{11379811877106581659740 2430375}{15845632502852867 5187087900672}x_1^{\frac{6889937869568708339499981931275}{2028240960365167042394725128601}}x_2^{\frac{7544423132589809964 0798371358899}{8112963841460668169578900514406}}
$$

$$
\oplus \frac{-19844302439616671184375517979487}{81129638414606681695789005144064}x_1^{\frac{10648909591190531843811749108 43}{8112963841460668169578900514406}}x_2^{\frac{6041092044511444312334136 1610413}{8112963841460668169578900514406}}
$$

$$
\oplus \frac{38420334371169026913494526372513}{81129638414606681695789005144064}x_1^{\frac{17362138752552321484255944932843}{8112963841460668169578900514406}}x_2^{\frac{93047833600821250929834168683413}{8112963841460668169578900514406}}
$$

$$
\oplus \frac{-17603947378294062525056571070551}{81129638414606681695789005144064}x_1^{\frac{6564683680998574221620489953367}{2028240960365167042394725128601}}x_2^{\frac{27428459948081432536048514362851}{8112963841460668169578900514406}}
$$

$$
\oplus \frac{40660689432491635572813473281449}{81129638414606681695789005144064}x_1^{\frac{8138995629356891296589182458867}{2028240960365167042394725128601}}x_2^{\frac{60065373103788240342541321435851}{8112963841460668169578900514406}}
$$

$$
\oplus \frac{-18724124908955366854716044525019}{40564819207303340847894502572032}x_1^{\frac{9606112199827178501273797702 1211}{8112963841460668169578900514406}}x_2^{\frac{4503206222300458382508431 1687365}{8112963841460668169578900514406}}
$$

$$
\oplus \frac{10408193496437482194218977650981}{40564819207303340847894502572032}x_1^{\frac{42358369791705053312612747043211}{8112963841460668169578900514406}}x_2^{\frac{77668975378711391631577118760365}{8112963841460668169578900514406}}
$$

$$
\oplus \frac{33780442675365993370749930 47453}{126765060022822940149670 3205376}x_1 x_2^{\frac{5818601932627716378583698 9899259}{8112963841460668169578900514406}}
$$

$$
\oplus \frac{44618467267239594276882973631597}{20282409603651670423947251286016}x_1^{\frac{99834969438155384051281539 3871}{8112963841460668169578900514406}}x_2^{\frac{08085321703760770307471696 92439}{25353012004564588029934 0641075}}
$$

$$
\oplus \frac{8303211062024807}{18014398509481984}x_1^{\frac{5315625921210391264531289425775}{2028240960365167042394725128601}}x_2^{\frac{4280731817019129183430556428 5899}{8112963841460668169578900514406}}
$$

$$
\oplus \frac{-8165645211761982742118 2154083}{20282409603651670423947251286016}x_1^{\frac{1124600062866011889863797 3096971}{8112963841460668169578900514406}}x_2^{\frac{6905895518496787064523625284043}{50706024009129176059868128 2150}}
$$

$$
\oplus \frac{64382118954969073630997358422927}{20282409603651670423947251286016}x_1^{\frac{6778740845753648164496920339037}{8112963841460668169578900514406}}x_2^{\frac{43070706318462417398441659587307}{8112963841460668169578900514406}}
$$

$$
\oplus \frac{13737969485405769628670110823819}{50706024009129176059868128 21504}x_1^{\frac{6690559447393050501252433933227}{2028240960365167042394725128601}}x_2^{\frac{4422357277763732412282131240381}{507060240091291760598681282150}}
$$

$$
\oplus \frac{19681995235611859526693202637247}{20282409603651670423947251286016}x_1^{\frac{80412445305952132226220780 42137}{8112963841460668169578900514406}}x_2^{\frac{27692005162376545446910233973947}{8112963841460668169578900514406}}
$$

$$
\left.\oplus \frac{25629385555566466102594071877399}{50706024009129176059868128 21504}x_1^{\frac{60030926843017208828918616 79501}{10141204801825835211973625 64300}}x_2^{\frac{85559422775418270765570856 9773}{25353012004564588029934 0641075}}\right)
$$

$$
\oslash\left(0\, x_1^{\frac{5315625921210391264531289425775}{2028240960365167042394725128601}}x_2^{\frac{4280731817019129183430556 4285899}{8112963841460668169578900514406}}\right.
$$

$$
\oplus \frac{11379811877106581659740 2430375}{15845632502852867 5187087900672}x_1^{\frac{6889937869568708339499981931275}{2028240960365167042394725128601}}x_2^{\frac{7544423132589809964 0798371358899}{8112963841460668169578900514406}}
$$

$$
\oplus \frac{-19844302439616671184375517979487}{81129638414606681695789005144064}x_1^{\frac{10648909591190531843811749108 43}{8112963841460668169578900514406}}x_2^{\frac{6041092044511444312334136 1610413}{8112963841460668169578900514406}}
$$

$$
\oplus \frac{38420334371169026913494526372513}{81129638414606681695789005144064}x_1^{\frac{17362138752552321484255944932843}{8112963841460668169578900514406}}x_2^{\frac{93047833600821250929834168683413}{8112963841460668169578900514406}}
$$

$$
\oplus \frac{-17603947378294062525056571070551}{81129638414606681695789005144064}x_1^{\frac{6564683680998574221620489953367}{2028240960365167042394725128601}}x_2^{\frac{27428459948081432536048514362851}{8112963841460668169578900514406}}
$$

$$
\oplus \frac{40660689432491635572813473281449}{81129638414606681695789005144064}x_1^{\frac{8138995629356891296589182458867}{2028240960365167042394725128601}}x_2^{\frac{60065373103788240342541321435851}{8112963841460668169578900514406}}
$$

$$
\oplus \frac{-18724124908955366854716044525019}{40564819207303340847894502572032}x_1^{\frac{9606112199827178501273797702 1211}{8112963841460668169578900514406}}x_2^{\frac{4503206222300458382508431 1687365}{8112963841460668169578900514406}}
$$

$$
\left.\oplus \frac{10408193496437482194218977650981}{40564819207303340847894502572032}x_1^{\frac{42358369791705053312612747043211}{8112963841460668169578900514406}}x_2^{\frac{77668975378711391631577118760365}{8112963841460668169578900514406}}\right).
$$

As we noted previously, redundant monomials may exist within this representation. Removing these gives the following reduced representation,

$$
\begin{aligned}
\Bigg(& \frac{8303211062024807}{18014398509481984} x_1^{\frac{5315625921210391264531289425775}{2028240960365167042394725128601}} x_2^{\frac{4280731817019129183430556428589}{8112963841460668169578900514406}} \\
\oplus\; & \frac{11379811877106581659740243037 5}{15845632502852867518708790067 2} x_1^{\frac{6889937869568708339499981931275}{2028240960365167042394725128601}} x_2^{\frac{7544423132589809964079837135889 9}{8112963841460668169578900514406}} \\
\oplus\; & \frac{-19844302439616671184375517979487}{8112963841460668169578900514064} x_1^{\frac{1064890959119053184381174910843}{8112963841460668169578900514406}} x_2^{\frac{6041092044511444312334136161041 3}{8112963841460668169578900514406}} \\
\oplus\; & \frac{3842033437116902691349452637251 3}{8112963841460668169578900514064} x_1^{\frac{1736213875255232148425594493284 3}{8112963841460668169578900514406}} x_2^{\frac{9304783360082125092983416868341 3}{8112963841460668169578900514406}} \\
\oplus\; & \frac{4066068943249163557281347328144 9}{8112963841460668169578900514064} x_1^{\frac{8138995629356891296589182458867}{2028240960365167042394725128601}} x_2^{\frac{6006537310378824034254132143585 1}{8112963841460668169578900514406}} \\
\oplus\; & \frac{10408193496437482194218977650981}{40564819207303340847894502572032} x_1^{\frac{4235836979170505331261274704321 1}{8112963841460668169578900514406}} x_2^{\frac{7766897537871139163157711876036 5}{8112963841460668169578900514406}} \\
\oplus\; & \frac{33780442675365993370749930474 53}{12676506002282294014967032053 76} x_1 x_2^{\frac{5818601932627716378583698989925 9}{8112963841460668169578900514406}} \\
\oplus\; & \frac{8303211062024807}{18014398509481984} x_1^{\frac{5315625921210391264531289425775}{2028240960365167042394725128601}} x_2^{\frac{4280731817019129183430556428589 9}{8112963841460668169578900514406}} \\
\oplus\; & \frac{6438211895496907363099735842292 7}{20282409603651670423947251286016} x_1^{\frac{6778740845753648164496920339037}{8112963841460668169578900514406}} x_2^{\frac{4307070631846241739844165958730 7}{8112963841460668169578900514406}} \\
\oplus\; & \frac{13737969485405769628670110823819}{50706024009129176059868128215 04} x_1^{\frac{6690559447393050501252433933227}{2028240960365167042394725128601}} x_2^{\frac{442235727776373241228213124038 1}{50706024009129176059868128215 0}} \Bigg) \\[6pt]
\oslash \Bigg(& 0 x_1^{\frac{5315625921210391264531289425775}{2028240960365167042394725128601}} x_2^{\frac{4280731817019129183430556428589 9}{8112963841460668169578900514406}} \\
\oplus\; & \frac{11379811877106581659740243037 5}{15845632502852867518708790067 2} x_1^{\frac{6889937869568708339499981931275}{2028240960365167042394725128601}} x_2^{\frac{7544423132589809964079837135889 9}{8112963841460668169578900514406}} \\
\oplus\; & \frac{-19844302439616671184375517979487}{8112963841460668169578900514064} x_1^{\frac{1064890959119053184381174910843}{8112963841460668169578900514406}} x_2^{\frac{6041092044511444312334136161041 3}{8112963841460668169578900514406}} \\
\oplus\; & \frac{3842033437116902691349452637251 3}{8112963841460668169578900514064} x_1^{\frac{1736213875255232148425594493284 3}{8112963841460668169578900514406}} x_2^{\frac{9304783360082125092983416868341 3}{8112963841460668169578900514406}} \\
\oplus\; & \frac{-17603947378294062525056571070551}{8112963841460668169578900514064} x_1^{\frac{6564683680998574221620489953367}{2028240960365167042394725128601}} x_2^{\frac{2742845994808143253604851436285 1}{8112963841460668169578900514406}} \\
\oplus\; & \frac{4066068943249163557281347328144 9}{8112963841460668169578900514064} x_1^{\frac{8138995629356891296589182458867}{2028240960365167042394725128601}} x_2^{\frac{6006537310378824034254132143585 1}{8112963841460668169578900514406}} \\
\oplus\; & \frac{10408193496437482194218977650981}{40564819207303340847894502572032} x_1^{\frac{4235836979170505331261274704321 1}{8112963841460668169578900514406}} x_2^{\frac{7766897537871139163157711876036 5}{8112963841460668169578900514406}} \Bigg).
\end{aligned}
$$

Since we have a 2-dimensional input domain, we can visualize the linear regions in which the neural network partitions the input domain, Figure 8a. By leveraging the capabilities of the OSCAR library, we can further compute geometric quantities of the linear regions, such as their volumes, Table 6. The extent to which these measurements are useful for applications such as interpretability is left for future work.

Note how linear regions may be constructed as unions of convex polyhedra to form non-convex regions, for instance in Figure 8a, we see that linear region 6 is constructed as the union of various convex polyhedra. Moreover, despite the same linear map acting on different polyhedra, these polyhedra may be disconnected and thus form separate linear regions, for instance in Figure 8a linear region 5 is acted on by the same linear map as linear region 6 but it is disconnected from linear region 6. Our Algorithm 1 accounts for both of these scenarios to ensure that an accurate enumeration of the linear regions is provided.

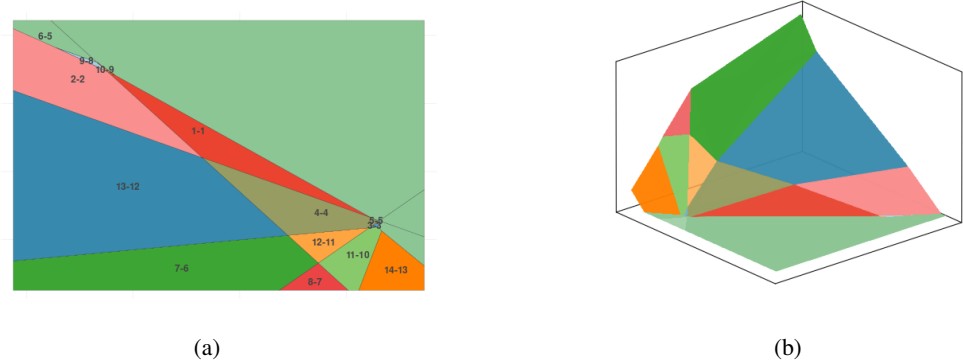

|         |         |
|:-------:|:-------:|
| (a)     | (b)     |

Figure 8: **Left:** The linear regions of a $[2, 6, 1]$ randomly initialized neural network. Each region is annotated first with its linear region number and secondly by an index identifying which linear map acts on that region, Table 5. **Right:** Visualization of the linear maps acting on these linear regions.

In Table 5, we explicitly identify the linear maps acting on the identified linear regions.

TABLES

| Architecture | Linear regions | Runtime (s) |
| --- | --- | --- |
| [2, 6, 1] | 11.84 | 2.57 |
| [3, 5, 1] | 20.88 | 4.76 |
| [4, 4, 1] | 14.2 | 1.05 |
| [5, 3, 1] | 7.4 | 0.35 |
| [6, 2, 1] | 4.0 | 0.167 |
| [3, 2, 2, 1] | 5.56 | 25.18 |
| [3, 3, 2, 1] | 14.72 | 38.51 |

(a) Symbolic calculation

| Architecture | Linear regions | Runtime (s) |
| --- | --- | --- |
| [2, 6, 1] | 16.84 | 0.214 |
| [3, 5, 1] | 20.8 | 0.217 |
| [4, 4, 1] | 14.4 | 0.217 |
| [5, 3, 1] | 7.96 | 0.205 |
| [6, 2, 1] | 4.0 | 0.197 |
| [3, 2, 2, 1] | 6.72 | 0.187 |
| [3, 3, 2, 1] | 12.12 | 0.172 |

(b) Numerical calculation, $N = 1000$ and $R = 5$

| Architecture | Linear regions | Runtime (s) |
| --- | --- | --- |
| [2, 6, 1] | 14.3 | 0.168 |
| [3, 5, 1] | 19.76 | 0.191 |
| [4, 4, 1] | 15.44 | 0.153 |
| [5, 3, 1] | 8.0 | 0.147 |
| [6, 2, 1] | 4.0 | 0.165 |
| [3, 2, 2, 1] | 5.96 | 0.176 |
| [3, 3, 2, 1] | 11.84 | 0.172 |

(c) Numerical calculation, $N = 1000$ and $R = 20$

| Architecture | Linear regions | Runtime (s) |
| --- | --- | --- |
| [2, 6, 1] | 18.6 | 0.849 |
| [3, 5, 1] | 21.56 | 0.903 |
| [4, 4, 1] | 14.84 | 1.031 |
| [5, 3, 1] | 7.96 | 0.971 |
| [6, 2, 1] | 4.0 | 0.743 |
| [3, 2, 2, 1] | 6.16 | 1.007 |
| [3, 3, 2, 1] | 12.92 | 0.969 |

(d) Numerical calculation, $N = 5000$ and $R = 5$

| Architecture | Linear regions | Runtime (s) |
| --- | --- | --- |
| [2, 6, 1] | 17.04 | 0.731 |
| [3, 5, 1] | 21.32 | 0.818 |
| [4, 4, 1] | 15.2 | 0.746 |
| [5, 3, 1] | 8.0 | 0.747 |
| [6, 2, 1] | 4.0 | 0.749 |
| [3, 2, 2, 1] | 6.04 | 0.912 |
| [3, 3, 2, 1] | 13.32 | 0.967 |

(e) Numerical calculation, $N = 5000$ and $R = 20$

Table 1: Comparison between numerical and symbolic calculations.

Table 2: Number of iterations in the PVZ algorithm and average time to solve LP during each iteration.

| # iterations | 94 | 86 | 67 | 83 | 99 | 86 | 75 | 83 |
| --- | --- | --- | --- | --- | --- | --- | --- | --- |
| Time per LP | 0.0042 | 0.0026 | 0.0025 | 0.0026 | 0.0025 | 0.0025 | 0.0026 | 0.0026 |

(a) $m_p = 10$, $m_q = 5$ and $n = 3$.

| # iterations | 2437 | 1110 | 1731 | 1441 | 1432 | 1706 | 1741 | 1095 |
| --- | --- | --- | --- | --- | --- | --- | --- | --- |
| Time per LP | 0.0152 | 0.0093 | 0.0092 | 0.0098 | 0.0098 | 0.0102 | 0.0095 | 0.0097 |

(b) $m_p = 15$, $m_q = 9$ and $n = 6$.

| # iterations | 2 | 607 | 525 | 80 | 194 | 355 | 78 | 19 |
| --- | --- | --- | --- | --- | --- | --- | --- | --- |
| Time per LP | 0.0027 | 0.0027 | 0.0026 | 0.0027 | 0.0032 | 0.0027 | 0.0028 | 0.0027 |

(c) $m_p = 15$, $m_q = 5$ and $n = 7$.

Table 3: Lower bounds and true values of Hoffman constants

| Sample | 1 | 2 | 3 | 4 | 5 | 6 | 7 | 8 |
|---|---|---|---|---|---|---|---|---|
| Lower bounds $H_L$ | 0.239 | 0.153 | 0.209 | 0.316 | 0.366 | 0.361 | 0.374 | 0.399 |
| Time $H_L$ | 0.206 | 0.205 | 0.204 | 0.206 | 0.206 | 0.207 | 0.211 | 0.216 |
| True values $H$ | 0.555 | 0.621 | 0.594 | 1.105 | 1.142 | 0.649 | 0.778 | 1.876 |
| Time $H$ | 0.644 | 0.686 | 0.651 | 0.638 | 0.674 | 0.638 | 0.657 | 0.676 |
| Upper bounds $H^U$ | 1.033 | 0.906 | 0.899 | 1.966 | 1.784 | 1.183 | 1.448 | 2.728 |
| Time $H^U$ | 0.001 | 0.001 | 0.002 | 0.001 | 0.001 | 0.002 | 0.001 | 0.001 |

(a) $m_p = 2$, $m_q = 3$ and $n = 6$

| Sample | 1 | 2 | 3 | 4 | 5 | 6 | 7 | 8 |
|---|---|---|---|---|---|---|---|---|
| Lower bounds $H_L$ | 0.214 | 0.271 | 0.237 | 0.222 | 0.323 | 0.145 | 0.159 | 0.371 |
| Time $H_L$ | 0.448 | 0.430 | 0.443 | 0.420 | 0.441 | 0.443 | 0.446 | 0.440 |
| True values $H$ | 0.970 | 0.901 | 1.045 | 0.555 | 1.023 | 1.402 | 0.530 | 0.843 |
| Time $H$ | 5.619 | 5.535 | 5.593 | 5.567 | 5.614 | 5.705 | 5.489 | 5.605 |
| Upper bounds $H^U$ | 1.426 | 1.437 | 2.129 | 1.058 | 2.328 | 2.607 | 1.208 | 1.748 |
| Time $H^U$ | 0.007 | 0.007 | 0.006 | 0.007 | 0.007 | 0.008 | 0.006 | 0.007 |

(b) $m_p = 3$, $m_q = 4$ and $n = 9$

| Sample | 1 | 2 | 3 | 4 | 5 | 6 | 7 | 8 |
|---|---|---|---|---|---|---|---|---|
| Lower bounds $H_L$ | 0.287 | 0.180 | 0.186 | 0.243 | 0.329 | 0.304 | 0.246 | 0.177 |
| Time $H_L$ | 0.708 | 0.693 | 0.745 | 0.749 | 0.719 | 0.701 | 0.710 | 0.687 |
| True values $H$ | 1.870 | 1.219 | 2.158 | 1.287 | 1.156 | 1.075 | 1.855 | 2.138 |
| Time $H$ | 36.456 | 36.089 | 37.885 | 37.785 | 36.299 | 36.562 | 35.724 | 33.566 |
| Upper bounds $H^U$ | 3.970 | 3.098 | 4.973 | 3.727 | 10.342 | 1.960 | 6.269 | 5.535 |
| Time $H^U$ | 0.086 | 0.085 | 0.084 | 0.050 | 0.052 | 0.051 | 0.084 | 0.083 |

(c) $m_p = 5$, $m_q = 4$ and $n = 8$

| Sample | 1 | 2 | 3 | 4 | 5 | 6 | 7 | 8 |
|---|---|---|---|---|---|---|---|---|
| Lower bounds $H_L$ | 0.194 | 0.229 | 0.246 | 0.194 | 0.190 | 0.216 | 0.199 | 0.231 |
| Time $H_L$ | 0.791 | 0.980 | 0.736 | 0.666 | 0.693 | 0.662 | 0.698 | 0.680 |
| True values $H$ | 1.079 | 0.768 | 0.932 | 0.797 | 0.895 | 0.826 | 0.672 | 0.985 |
| Time $H$ | 91.833 | 95.885 | 71.201 | 69.635 | 69.700 | 69.030 | 69.494 | 69.137 |
| Upper bounds $H^U$ | 3.280 | 1.679 | 2.711 | 4.417 | 6.425 | 2.642 | 2.359 | 2.016 |
| Time $H^U$ | 0.295 | 0.176 | 0.160 | 0.160 | 0.129 | 0.190 | 0.159 | 0.128 |

(d) $m_p = 7$, $m_q = 3$ and $n = 12$

| Experiment | Compute time |
|---|---|
| Redundant Monomials | 52.93 minutes |
| At the MNIST-level | 2.68 minutes |
| Table 1, symbolic calculations | 29.06 minutes |
| Table 1, numerical calculations | 6.13 minutes |
| Linear regions of invariant neural networks, Figure 6 | 24.63 minutes |

Table 4: Compute details.

| | Linear Map |
|---|---|
| 1 | $\dfrac{17777449875117333065930502866479}{81129638414606681695789005144064} + (x_1 \quad x_2)\begin{pmatrix} -\dfrac{117362138752552321484255944932843}{81129638414606681695789005144064} \\ -\dfrac{6743090713727204357199858939207}{40564819207303340847894502572032} \end{pmatrix}$ |
| 2 | $\dfrac{236039135561959028757175073016479}{81129638414606681695789005144064} + (x_1 \quad x_2)\begin{pmatrix} -\dfrac{7106489095119053184381174910843}{81129638414606681695789005144064} \\ -\dfrac{511124505594186396668752185855577}{40564819207303340847894502572032} \end{pmatrix}$ |
| 3 | $\dfrac{175534143689850721999986081755543}{81129638414606681695789005144064} + (x_1 \quad x_2)\begin{pmatrix} -\dfrac{581338995629356891296589182458867}{20282409603651670423947251286016} \\ -\dfrac{1174596110944422847940207210 37}{5070602400912917605986812821504} \end{pmatrix}$ |
| 4 | $\dfrac{2467659317368072804295773604453}{1267650600228229401496703205376} + (x_1 \quad x_2)\begin{pmatrix} -\dfrac{26889937869568708339499981931275}{20282409603651670423947251286016} \\ -\dfrac{14657276499952616981870172682455}{10141204801825835211973625643008} \end{pmatrix}$ |
| 5 | $0 + (x_1 \quad x_2)\begin{pmatrix} 0 \\ 0 \end{pmatrix}$ |
| 6 | $\dfrac{64382118954969073630997358422927}{20282409603651670423947251286016} + (x_1 \quad x_2)\begin{pmatrix} -\dfrac{44483762839087916893628237364063}{81129638414606681695789005144064} \\ -\dfrac{6241769120366527326120747021831}{2535301200456458802993406410752} \end{pmatrix}$ |
| 7 | $\dfrac{275132423198170357049046004762259}{81129638414606681695789005144064} + (x_1 \quad x_2)\begin{pmatrix} -\dfrac{169479993878240648721985039474431}{81129638414606681695789005144064} \\ -\dfrac{10544719203702376892200856846943}{10141204801825835211973625643008} \end{pmatrix}$ |
| 8 | $\dfrac{57238640684530172339958449873759}{81129638414606681695789005144064} + (x_1 \quad x_2)\begin{pmatrix} -\dfrac{980238727427488126256017207743}{81129638414606681695789005144064} \\ -\dfrac{8801801137461575644517898662257}{40564819207303340847894502572032} \end{pmatrix}$ |
| 9 | $-\dfrac{1025996126255525757911594478241}{81129638414606681695789005144064} + (x_1 \quad x_2)\begin{pmatrix} -\dfrac{56099635067710756426130787229743}{81129638414606681695789005144064} \\ -\dfrac{25120257715314979547764302198757}{40564819207303340847894502572032} \end{pmatrix}$ |
| 10 | $\dfrac{216867786387384658951175960410259}{81129638414606681695789005144064} + (x_1 \quad x_2)\begin{pmatrix} -\dfrac{215777241671673917021859809496431}{81129638414606681695789005144064} \\ -\dfrac{3656083337041431967003114432767}{2535301200456458802993406410752} \end{pmatrix}$ |
| 11 | $\dfrac{49815959752272649106529847334927}{20282409603651670423947251286016} + (x_1 \quad x_2)\begin{pmatrix} -\dfrac{90781010632521185193503007386063}{81129638414606681695789005144064} \\ -\dfrac{29046690625929460280294588971449}{10141204801825835211973625643008} \end{pmatrix}$ |
| 12 | $\dfrac{3378044267536599337074993047453}{1267650600228229401496703205376} + (x_1 \quad x_2)\begin{pmatrix} -\dfrac{15315625921210391264531289425775}{20282409603651670423947251286016} \\ -\dfrac{5288831177744633003029285899165}{5070602400912917605986812821504} \end{pmatrix}$ |
| 13 | $\dfrac{179146822334000678485908299899655}{81129638414606681695789005144064} + (x_1 \quad x_2)\begin{pmatrix} -\dfrac{5181054522745480099417093565705}{2535301200456458802993406410752} \\ -\dfrac{129307656659568521746027221589755}{81129638414606681695789005144064} \end{pmatrix}$ |

Table 5: The linear maps referenced in the annotations of Figure 8a.

| Linear Region | Volume |
|---|---|
| 1 | 3.9158 |
| 2 | Unbounded |
| 3 | 0.0196 |
| 4 | 3.8078 |
| 5 | 0.0043 |
| 6 | Unbounded |
| 7 | Unbounded |
| 8 | Unbounded |
| 9 | 0.1942 |
| 10 | 0.029 |
| 11 | 1.7305 |
| 12 | 1.0766 |
| 13 | Unbounded |
| 14 | Unbounded |

Table 6: Volumes of the linear regions identified in Figure 8a.

