# OpenReview forum: "Tropical Expressivity of Neural Networks"
_ICLR.cc/2025/Conference — Submitted to ICLR 2025_

### Official Review · Reviewer_3Zin · 2024-10-17

**Soundness:** 3
**Presentation:** 3
**Contribution:** 2
**Rating:** 5
**Confidence:** 4

**Summary:**

The authors study some theoretical and computational aspects of neural networks with piecewise linear activation functions. In Section 2, they are interested in the number of linear regions of such a network. They define the minimal effective radius around an input point as the smallest radius such that a ball of that radius intersects all the linear regions. They bound that radius from above by a Hoffman constant associated with the network. The main advertised application of that is that one can in principle enumerate all linear regions by sampling dense enough inside a ball of their estimated radius.

In Section 3, the authors consider networks that are invariant under a given group action. They provide lower and upper bounds on the number of linear regions by making use of the fundamental domain of the group action. Since the fundamental domain gives a tiling of the input space, the authors further explain that sampling inside the fundamental domain suffices to enumerate all linear regions.

In Section 4, the authors explain their software contribution. They provide a symbolic algorithm that finds all linear regions of a network once one has a representation of the network as a tropical rational function. The authors also suggest to measure the expressivity of a network by the number of monomial required in a tropical rational function representation. They provide an algorithm that reduces unnecessary monomials in a given representation.

The authors conclude with some experiments and discussion, but their main contributions are the theoretical contributions in Sections 2 and 3, and their software contribution described in Section 4.

**Strengths:**

Originality: The definition of the Hoffman constant and its usage for sampling linear regions is novel, to the best of my knowledge. Also, I believe that the monomial count expressivity measure has not been studied before.

Significance: I think it will be very helpful for future research that the authors made their software available within the open-source OSCAR library. The proposed ideas of the Hoffman constant and the fundamental region might play a significant role in the future understanding of finding all linear regions of a piecewise linear network.

Quality and clarity: The paper is well-written. The mathematical statements are sound and the experiments make sense.

**Weaknesses:**

My main concern is whether the paper's contributions are strong enough, which consist mainly of the author's software and their idea of the Hoffman constant to find a bounded region in the input space that intersects all linear regions. Making use of the fundamental domain of a given group action is a rather folklore idea, and regarding the monomial count expressivity measure I have some questions (see below). Moreover, and also very important: The paper oversells its contributions. I strongly suggest to tone down the description of the contributions and focus on describing what exactly is done instead.

1) For instance, the last sentence of the abstract is way too bold. The authors claim to provide "the foundations" but it is rather a small piece of the puzzle.

2)  Another example is the sentence "We study more than simply the number of linear regions, we provide further insight on their geometry". The only new geometric aspect I can see is the definition of the minimal effective radius. But that does not really give us new insights about the geometric properties of the regions, such as their size/volume, their number of facets or vertices, or any other information about the combinatorial type of the polytopes. I suggest to remove the sentence in question completely.

3) Similarly, in line 088, the authors write that the Hoffman constant "effectively gives a geometric characterization". I do not agree with that half-sentence, since the constant does not tell us (almost) anything about the geometric properties of the linear regions; it only tells us how "far away" the regions are at most located. If the authors have something else in mind, I suggest to replace the vague and way too broad statement "geometric characterization" by the exact properties the Hoffman constant tells us.

4) I also think that the phrase "guarantee for the linear regions" in line 088 is too strong. Please be more specific on what that guarantee means. The authors provide a largest region for where one has to sample. But if the sample is not dense enough, one will still miss linear regions (as the authors note in the appendix).

5) It is always hard to claim to have the "first work" as in line 531. Symbolic experiments on other deep learning architectures have for instance been done in "Pure and Spurious Critical Points: a Geometric Study of Linear Networks" by Trager, Kohn, Bruna or in "Geometry of Linear Convolutional Networks" by Kohn, Merkh, Montúfar, Trager.

Finally, here some smaller comments:

- Please add "piecewise" before "linear" in the first line of the abstract.

- I think it is inaccurate to call tropical geometry a "reinterpretation" of algebraic geometry. It is rather a "discrete version" of algebraic geometry.

- In line 139.5, remove one of the 2 words "its the"

- In lines 176 and 177, the row vectors a and a' should have an index i. That is, they should be a_i resp. a'_i.

- In line 191, it would be better to spell out "for all" before "U" to increase clarity.

- In line 205, should m_p and m_q be equal?

- In line 212, insert "in" after "x"

- typo in caption of Figure 3

**Questions:**

1) In Theorem 3.3, what happens if the group is infinite? Shouldn't your theorem imply that the number of linear regions is infinite in this case? Or will it also be true that the set U_c is empty such that your lower bound becomes trivially equal to 0?

2) A priory, more monomials does not necessarily mean more expressivity, since the monomials could satisfy some internal (algebraic) relations/constraints. For instance, in classical algebra, the polynomials that are large powers of linear terms (i.e., that are of the form (a*x+b)^N for some fixed large N) have many monomials but they can only express a small portion (namely, a 2-dimensional subset) of the function space. Does this behavior happen for the tropical representations of ReLU neural networks? If yes, then a more accurate measure of expressivity would be the dimension of the semi-algebraic set that consists of all tropical rational functions that a fixed network architectures parametrizes, as studied in the article "Functional dimension of feedforward ReLU neural networks" by Grigsby, Lindsey, Meyerhoff, Wu.

3) I don't understand what is meant with the number of monomials in Figure 9, since both the numerator and denominator have monomials. Do you plot the sum of the number of monomials of both numerator and denominator? Wouldn't in be better to also plot the product to get an idea of how far the upper bound in Example E.2 is?

---

> ### Author Response · Authors · 2024-11-28
> **Response to Reviewer Comments**
>
> We thank the reviewer for their careful reading of our work and would like to respond to the concerns raised and respond to the weaknesses raised and their questions.
>
> ## The geometric characterization of the Hoffman constant
>
> To clarify what we mean by "geometric characterization": the Hoffman constant gives us the radius of a ball centered at the origin which would intersect every linear region in the domain. We believe the word "geometric" is a fair description, although we acknowledge that geometry is a broad area and there are many other concepts and measurements that may be also considered geometric.
>
> ## Guarantee for the linear regions
>
> In response to the comment that "guarantee for the linear regions" in line 088 is too vague: we thank the reviewer for this point and will clarify it in the revision. When we mention a guarantee, we mean that in the limit if enough points were sampled, with probability 1 all the linear regions would eventually be found. We will clarify this in the revision.
>
> ## Previous work on symbolic approaches to deep learning
>
> The work "Pure and Spurious Critical Points: a Geometric Study of Linear Networks" by Trager, Kohn, Bruna and "Geometry of Linear Convolutional Networks" by Kohn, Merkh, Montúfar, Trager are mentioned as previous examples of work using symbolic approaches to deep learning. We thank the reviewer for providing these resources. Having reviewed them, we do believe our current statement on line 531 is potentially discrediting existing work: to clarify, we mean that ours is the first to leverage tropical symbolic computation to perform experiments on deep neural networks in a practical setting, which we believe is the case, and which we will clarify in the revision.

---

> ### Author Response · Authors · 2024-11-28
> **Response to Questions**
>
> ## Question 1
> In theorem 3.3, we are assuming that the group is finite. We apologize for the confusion, and will clarify this in our revision.
>
> ## Question 2
> It may well be the case that the tropical rational maps we get from neural networks conform to algebraic constraints that are currently unknown. However, we remove redundant monomials for the purpose of our analysis, and we only claim that more *irredundant* monomials means more expressivity. Monomials are redundant if they are never the maximum linear term in either the numerator or denominator for any point in the input space. This means that we only look at monomials which contribute to adding complexity to the function that the network represents. In other words, if there were some algebraic constraint which meant that some monomials will always be redundant, then our symbolic analysis would simply ignore them. What we are measuring is therefore a legitimate estimation of how complicated the neural network is, and it is to the best of our knowledge a novel metric.
>
> ## Question 3
> In Figure 9 we are indeed plotting the sum of the number of monomials from both the numerator and denominator for each sample on the $x$-axis. We thank the reviewer for the suggestion to plot the product and will consider this addition for future work.

---

### Official Review · Reviewer_5XwZ · 2024-11-01

**Soundness:** 3
**Presentation:** 2
**Contribution:** 2
**Rating:** 3
**Confidence:** 4

**Summary:**

The paper investigates counting the linear regions of neural networks through tropical geometry. The authors relate the Hoffman constant of a tropical rational map to the minimal effective radius, which is the smallest radius $r$ such that a ball of radius $r$ intersects all linear regions. They further show that for an invariant network (with respect to a group $G$), one can sample from the fundamental domain of $G$ to obtain an upper bound on the number of linear regions. Lastly, the authors introduce an algorithm using symbolic representation to compute the number of linear regions and propose a new measure of complexity for ReLU neural networks, termed the “monomial count.”

**Strengths:**

The number of linear regions is a standard measure of complexity for neural networks with piecewise linear activation functions, and the authors provide new insights into computing this count for a given network. Specifically, they relate the minimal effective radius to the Hoffman constant of a tropical Puiseux rational map. Additionally, for invariant neural networks, their observation allows a significant reduction in the number of samples required to estimate the number of linear regions.

**Weaknesses:**

Overall, the contributions of the paper do not appear substantial enough, leading me to recommend rejection. The theoretical depth is limited, and I see few immediate practical benefits from the results, see also the questions below.
-  While relating the Hoffman constant to the radius of the ball may have theoretical appeal, I do not see its practical benefit, as it is NP-hard to compute the Hoffman constant. The appendix discusses computing the Hoffman constant of a matrix, but there is no discussion about the Hoffman constant for a rational Puiseux function, which is what relates to the radius of the ball.
- The result concerning sampling from the fundamental domain of the group action for an invariant network appears to be a straightforward observation.
- The algorithm for determining linear regions using Puiseux polynomials does not seem to offer clear advantages over existing methods. The paper states, “Our tools consider the full input domain of these neural networks and provide an exact geometric characterization of the linear regions. This now makes previously inaccessible avenues available for analyzing the geometry of linear regions of networks (see Appendix I)." However, algorithms for obtaining the entire polyhedral complex of neural networks already exist (e.g., Masden, 2022, https://arxiv.org/abs/2207.07696).

**Questions:**

- What is the benefit of relating the Hoffman constant to the radius of the ball? Are there any approximations for the Hoffman constant of a rational Puiseux function? Has the Hoffman constant been applied to Puiseux polynomials (and/or rational functions) in prior work?

- Assuming we know the Hoffman constant, are there theoretical guarantees that using it would lead to better sampling to estimate the number of linear regions?

- Is the definition of the Hoffman constant for a Puiseux rational map well-defined (Definition 2.2)? How do you deal with the fact that there can be infinitely many quotients?

- In the introduction, the paper mentions, “Our library opens the door for the extensive theory and existing software on symbolic computation and computational tropical geometry to be used to study neural networks." Could you provide an example of a property of a function computed by a neural network that is now accessible through symbolic computation but was previously inaccessible? A more detailed explanation of these advantages would be helpful.

- Why is the native tropical representation for neural networks useful for measuring complexity? Can you comment on how this representation compares to the best representation as a tropical rational function in terms of monomial complexity?

---

> ### Author Response · Authors · 2024-11-23
> **Response to Reviewer Comments**
>
> We thank the reviewer for their careful reading of our work and would like to respond to the concerns raised and respond to the weaknesses raised and their questions.
>
> ## Concerning the Hoffman constant
>
> On the one hand, we agree with the concern of the practical utility of the Hoffman constant since computing it exactly is NP-hard. Consequently, we utilize Theorem F.4 to construct an algorithm to obtain an upper bound on the constant, and thus an upper bound on the effective radius. In the paper, we consider the Hoffman constant computation of a tropical rational map, note how we consider different values of $m_p$, $m_q$, and $n$ which correspond to those in Definition 2.2.
>
> ## Concerning the fundamental domain
>
> Theorem 3.3 formalizes the intuitive notion that the linear regions of symmetric neural networks also have some underlying symmetry. Although it may seem like a straightforward observation, we show that its utility in facilitating the numerical sampling of neural network linear regions is significant.
>
> ## Concerning the algorithm for determining linear regions using Puiseux polynomials
>
> Existing methods such as “SplineCam: Exact Visualization and Characterization of Deep Network Geometry and Decision Boundaries” (which we do not currently reference but will in our revision) and the method mentioned by the reviewer “Algorithmic Determination of the Combinatorial Structure of the Linear Regions of ReLU Neural Networks” (again, which we do not currently reference but will in our revision) also characterize the linear regions of neural networks. Our method differs in that we get access to the polyhedral representations of the regions constituting the linear regions of the neural networks. This grants us the capability to explore geometrica questions such as the tracking of the volume of specific linear regions through training.

---

> > ### Author Response · Authors · 2024-11-23
> > **Response to Questions**
> >
> > ## Question 1
> >
> > The first part of this question will be answered within the response to the second question: We provide methods for computing lower and upper bounds on the Hoffman constant with Algorithms 5 and 6 respectively. To the best of our knowledge, there is no existing work that extends the Hoffman constant to tropical polynomials or rational maps, as we do here.
> >
> > ## Question 2
> >
> > The benefit of relating the Hoffman constant to the minimal effective radius is that we gain a guarantee that within a ball of this radius, all of the linear regions of the neural network are present. Therefore, we can sample points within this ball, without worrying that we are potentially missing linear regions due to them being outside our sampling domain.
> >
> > ## Question 3
> >
> > Definition 2.2 is specific to the algebraic representation of the tropical rational map being considered. However, we note that multiple algebraic expressions may correspond to the same tropical rational map when viewed as a function from its input domain to its output domain. Therefore, to ensure our definition is well-defined we take the Hoffman constant to be the minimum over all these possible algebraic realizations of this tropical rational map, as noted on lines 181 and 182.
> >
> > ## Question 4
> >
> > One example would be calculating the volume of the linear regions. Although previous works such as “SplineCam: Exact Visualization and Characterization of Deep Network Geometry and Decision Boundaries” compute average region volume statistics, they do not provide the ability to compute the volumes of individual regions. We demonstrate this capability in our analysis of the neural network trained on MNIST and in Table 6, however, we agree that we were not explicit enough in demonstrating these previously inaccessible capabilities. Furthermore, it is also possible to use the algebraic expression of the neural network as a tropical rational map to explore algebraic questions.
> >
> > ## Question 5
> >
> > The native tropical representation is obtained by implementing the construction outlined in the proof starting on 872. A priori we have reason to doubt that counting the number of monomials in this expression is a useful measure of complexity, due to the issue of redundant monomials we outline in the paragraph titled “Pruning Tropical Expressions” starting on line 394. However, we have reasons to believe that the number of monomials in the pruned tropical representations are indicative of complexity as they can be related to the linear regions of the neural network, which is an established measure of neural network expressivity. Pruning these tropical expressions is expensive as it requires combinatorially many computations in the number of monomials. Therefore, for some experiments, we use the native monomial counts instead. We here justify this with the observation that the number of monomials in the native expression and the pruned expression have an approximate linear relationship, therefore, conclusions made on the native monomials counts should hold when redundant monomials are pruned. We agree that this could have been more explicit and we will address this in a revision.

---

> > > ### Comment · Reviewer_5XwZ · 2024-11-25
> > >
> > > I thank the authors for the detailed response.
> > >
> > > **Regarding question 2 and 3**
> > >  The algorithms compute the Hoffman constant of a single matrix, right? Since for a tropical Puiseux rational map f, there can be infinitely many p and q  such that $f=p \oslash q$, I still do not see how you can compute (neither define properly) the Hoffman constant of f. Could you explain how you compute the constant and why the minimum is not an infimum?
> > >
> > > **Regarding question 5**
> > > I apologize if my questions was not clear. Here is what I meant: Let $f$ be the function computed by the neural network. Let $p \oslash q$ be the representation for the function induced by the neural network, as you explained (you can also consider this representation pruned).  Now, let $p^* \oslash q^*$ be a representation of $f$ that is minimal wrt to monomial complexity. How does the monomial complexity of $p \oslash q$ relate to $p^* \oslash q^*$? If there are no bounds on that relationship, could you explain why you consider the monomial complexity of the (pruned) native tropical representation a good complexity measure for the function $f$?
> > >
> > > I still have concerncs about the depth of the contribution and I will keep my score for now.

---

> > > > ### Author Response · Authors · 2024-12-03
> > > > **Response to follow-up comments and quetsions**
> > > >
> > > > ## Regarding Questions 2 and 3
> > > >
> > > > The algorithms do compute the Hoffman constant of a single matrix. From eq(4), we compute the Hoffman constant of a tropical rational map as a maximum over Hoffman constants of matrices, so we can just apply our algorithms on the matrices individually and then take the maximum to get the Hoffman constant of the tropical rational map. For our computations, we just perform this computation over the pruned representation of the tropical rational map, as provided by Algorithm 2. This provides an upper bound on the Hoffman constant of the tropical rational map since we do not rule out the possibility of a different representation leading to a lower Hoffman constant. We should have been more explicit about this in the paper. In lines 184/185 we do say that the Hoffman constant of $p\oslash q$ is taken to be the *minimal* Hoffman constant over all possible expressions. We acknowledge that this is ambiguous as to whether we are taking the minimum or infimum. We would like to clarify that we are taking the infimum here as we have provided no theoretical justification that we could take the minimum. However, we emphasize that this infimum is still a well-defined quantity as Hoffman constants are always positive so the set of Hoffman constants from different representations of the tropical rational map is bounded from below.
> > > >
> > > > ## Regarding Question 5
> > > >
> > > > We currently have no theoretical connection between the monomial complexity of the pruned representation $p\oslash q$ and the representation $p^*\oslash q^*$ achieving the minimal monomial complexity. However, we expect monomial complexity of the pruned representation $p\oslash q$ to be a useful measure of complexity since each of its monomial nontrivially contributes to the linear regions of $f$ as per our definition of a monomial not being redundant. Therefore, as linear regions have been established as useful measure of complexity, we expect that the monomial complexity of the pruned representation will also be a good complexity measure for $f$.

---

### Official Review · Reviewer_64tt · 2024-11-03

**Soundness:** 3
**Presentation:** 1
**Contribution:** 2
**Rating:** 3
**Confidence:** 4

**Summary:**

This paper studies the expressivity of neural networks through a tropical lens. Through a suite of algebraic tools, the authors characterize the linear regions of a neural network using the Hoffman constant, discuss sampling improvements for invariant neural networks through the fundamental domain, and provide a symbolic framework for computing linear regions of neural networks.

**Strengths:**

The paper tackles important problems that can aid our understanding of deep networks. Tropical geometry has been getting increasingly popular in recent years, and has been the tool of choice for geometric analyses using polyhedral geometry. The introduction of the manuscript is crisp and easy to follow, introducing the main subject areas and discussing the contributions clearly. Finally, there is a significant number of contributions.

**Weaknesses:**

The weaknesses of the work can be summarized in the following axes:

- a severe lack of polish after the introduction,
- confusing results, mainly due to the number of contributions,
- it is unclear what the benefits of the contributions are, either at a theoretical or a empirical level.

**Lack of polish**

All main sections of the paper are introduced very abruptly, and the defining concepts of each of them are delegated to the appendices. This makes the paper extremely difficult to read, and is currently not self-contained without the appendices. Examples of this is the complete omission of a discussion on tropical polynomials, the discussion of Lemma A.19 or Equation 9 without their exposition, and all algorithms being discussed, but never presented, to name a few.

There are many typos and there is a lack of explanations at multiple points in the paper. An example of this lack are Figures 2 and 3, where, beyond the typos, Figure 2 lacks any real caption and what is portrayed in these figures, what the colors mean, or why there is a different number of lines on Fig 2(a) and Fig 2(b) is not clear.

Overall while the three different contributions are connected, the connection is not emphasized and the presentation really hinders the readability and the flow of the work. By trying to include all of them in the same work, it becomes extremely difficult to present each contribution in a clear and rigorous manner, especially at such a page limit. I believe the work could greatly benefit from either splitting the contributions, or investing a significant effort in crafting a coherent thread that connects the contributions, most likely at a venue with a much higher page limit.

**Confusing results**

In terms of the Hoffman constant, it seems that the authors compute the constant, then use it to find the radius of a ball that is needed to intersect all the linear regions. However, if I’m understanding things correctly, this gives no estimate on the number of linear regions. Essentially we have to compute the Hoffman constant (which is NP-hard), and then have to sample many points in that radius to get a probabilistic statement about the number of regions given a sampling size, and even then I don’t see any statements of that effect. Most works I’m aware of on counting the number of linear regions do it without having access to a trained network, and are statements about the architecture, not the specific network, so I’m unsure exactly what the main contribution is here.

Section 4.1 is also confusing and the contribution is again unclear. For example, in line 359 it is stated that we can detect when some intersections may be empty. The way this is stated implies that we have a way of checking if intersections are empty, but not a way to find such intersections. That seems to actually be the case, since on 377 a set of indices I is assumed instead of being computed. A similar comment as above can be made, that this works for specific networks and does not characterize the network class. Also, how are we computing these regions? Are we combinatorially testing all combinations?

**Experimental results**

Simply put, the experimental section is very rushed and the results are not convincing (more in the next section). Starting of with the characterization of width versus depth, what exactly is Figure 4 conveying? There is no comparison between depth versus width, for the same number of neurons. The fact that expressivity increases when we increase the number of nodes is not surprising. A minor comment is also that the range of $k$ doesn’t correctly correspond to Figure 4.

For Figure 5, the trained and untrained networks seem to hardly have any difference in their pruning rate, and the size of the network is so small where no trends could be observed.

**Questions:**

In terms of questions:

- Early in the introduction the authors mention that expressivity is one of the most important approaches to measure performance. Are the authors aware of any references to support that claim? Usually overly complex networks (i.e., very expressive ones) tend to overfit the data and lead to poor performance, exactly the opposite of what the authors are arguing.
- Given that the purpose of the paper is to aid our understanding of deep networks, there seems to be a limited intuitive explanation of the methods that are used. For example, what is an intuitive explanation of the Hoffman constant? Section 2.1 makes an attempt at explaining, but actually describes the distance $d(u, P(A, B))$ and not the constant, and does so by restating the equation instead of an intuitive explanation.
- About the fundamental domain:
    - It seems that things only work for discrete groups, if so that should be a concrete discussion.
    - There is mention of groups acting on vector spaces, but no discussion about group representations (or at least how these groups act on the vector spaces) is presented.
    - The authors state that in the specific example of 3.2, a factorial improvement is made. However, clearly, $\Delta$ is an infinite set. There is no asymptotic analysis as to how many samples we need from $\Delta$, so how are you claiming an improvement without a sampling statement when sampling from an infinite set?
- Line 395 is very imprecise. What does information mean here? The expression and the function both encode the same function.
- On 423, the authors argue their use of small networks saying that they help highlight intuitive insights that can aid understanding. What are the insights that are gained in doing so? Of the three presented experiments, the first one has no comparison on depth vs width (which is the titular motivation for the section), the second one shows a marginal change which is unclear if it would persist when the layers/nodes increase, and the final one has no takeaways listed.

---

> ### Author Response · Authors · 2024-11-22
> **Response to Reviewer Comments**
>
> We thank the reviewer for their careful reading of our work and would like to respond to the concerns raised and respond to the weaknesses raised and their questions.
>
> ## Lack of polish
>
> We acknowledge the reviewer’s concern about the reliance on appendices and abrupt section transitions. These are largely due to the conference format, where comprehensive theoretical development often necessitates lengthy appendices. This practice is common in many ML papers, but we agree that incorporating key definitions and discussions (e.g., tropical polynomials, Lemma A.19, and Equation 9) into the main text would improve readability. We will address this in the revision.
>
> Regarding Figures 2 and 3, we acknowledge the lack of clear captions and explanations, as well as the typos throughout the manuscript.
> We sincerely apologize and will fix these issues in the revised version.
>
> Finally, while the three contributions are tightly interconnected, we appreciate the suggestion to consider splitting them or submitting to a venue with a higher page limit. However, we believe the contributions form a cohesive framework and will refine the manuscript to better emphasize their connections and improve clarity.
>
> ## Confusing results
>
> Indeed, the Hoffman constant provides no estimate on the number of linear regions of the neural network, however, for sampling-based techniques for counting the number of linear regions it provides a guarantee that the sampling domain contains all of the linear regions. The other works familiar to the reviewer with typically consider the maximum number of linear regions that a particular architecture can represent. Our contribution, outlined in Section 4,  are that given a neural network, we can compute exactly the number of linear regions it has. This is useful, for instance, in the context of studying this number empirically (see e.g., https://proceedings.mlr.press/v97/hanin19a.html).
>
> Given a pair of polyhedra, we can compute their intersection (see Lemma A.4. in the appendix for a formula!) and we can check whether this intersection is empty. Therefore, on line 377 we assume we have the set of indices I since in practice we can compute this set of indices. We do these computations on the specific neural network being considered as we want to compute the exact number of linear regions this neural network has. In this work, we do not consider the linear regions of the neural network class. Indeed, to compute the set I we first collect all the pairs (i,j) which are operated on by a specific linear map, and the we determine which pairs of these pairs satisfy (i) and (iii) of (★). All of this is done computationally by leveraging the OSCAR package.
>
> ## Experimental results
>
> We acknowledge the reviewer's observation that our experimental section is considered to be large scale. However, the primary focus of our paper is on developing theoretical tools for understanding neural network expressivity using tropical geometry. Our algorithms and software are grounded in this theoretical framework, operationalizing our insights into practical methods for exact expressivity analysis. They are integral to demonstrating the utility of our theoretical contributions, rather than standalone practical developments.
> The experiments serve as proof-of-concept demonstrations to illustrate these theoretical contributions. Regarding the reviewer's comment on width-depth comparisons, we have conducted additional calculations to explore this relationship further, as we agree it provides valuable insight. These results will be incorporated into the revised manuscript to strengthen our experimental findings.
> Lastly, we appreciate the reviewer’s observation regarding the range of $k$ in Figure 4 and will address this discrepancy in the revised manuscript for clarity.

---

> > ### Author Response · Authors · 2024-11-23
> > **Response to Questions**
> >
> > ## Question 1
> >
> > There are various notions of performance associated with a neural network/neural network architecture. The notion of performance we are interested in here would be that of the flexibility of a given architecture, i.e., its expressivity (how well it can fit a given dataset), while the notion of performance that the reviewer seems to have in mind seems to correspond to the generalization capabilities of a given neural network. We agree with the reviewer that these should not be confused, and will update lines 36-37 in our revision to clarify that we do not mean to conflate these. The use of linear regions as a tool for measuring the expressivity (in connection to performance) is well established in the literature (see e.g., the discussion in https://arxiv.org/pdf/1606.05336).
> >
> > ## Question 2
> >
> > Due to the space constraints, we choose to focus on the contributions of this paper rather than providing in-depth discussion and intuition for all the mathematical tools used (though an attempt is made in the appendix). That being said, we do understand that this makes the mathematical content harder to grasp, and we are happy to add more discussion in the appendix. To answer the reviewer’s question, here is an intuitive explanation of the Hoffman constant:
> > For a given matrix $A$ and a vector $b$, we can consider the polyhedron $P(A, b) $ and try to understand how the distance $d(x, P(A, b))$ changes as we vary $x$. For $x$ not in the polyhedron, we must have some indices $i$ such that $(Ax)\_i > b\_i$.  We can measure the failure of $x$ to satisfy the inequality $Ax \le b$ by considering the size of the vector $(Ax-b)\_+$ (notice that this is “truncating” all the coordinates that do satisfy the system). Naively, the hope is to understand the distance from $x$ to the polyhedron in terms of how how "badly" $x$ fails to lie in the polyhedron, i.e., how large this error term $\||(Ax-b)\_+\||$ is. Surprisingly, this turns out to be possible, at the cost of multiplying the error term by some constant which (even more surprisingly!) depends only on the matrix $A$. This constant is the Hoffman constant.
> >
> > ## Question 3
> >
> > The results in the fundamental domain section are indeed only valid for finite groups; this is an omission that we will fix in our revision. One point worth clarifying is that we are working with group actions on $\mathbb{R}^n$ as a set rather than a vector space, as defined [here](https://en.wikipedia.org/wiki/Group_action), and then specialize to the permutation action of the symmetric group on $\mathbb{R}^n$, which is discussed in detail in the appendix. Concerning our claim about sample efficiency: when using the sampling method to compute the number of linear regions, we usually restrict to a ball (or hypercube) around the origin, and sample a number of points proportional to the volume of the region we are considering. Our factorial improvement claim relates to the fact that the volume of $ [-R, R]$ is $n!$ times larger than the volume of $ [-R, R]  \cap \Delta$, meaning that we can reduce the number of sample points by a factor of $n!$.
> >
> > ## Question 4
> >
> > What we mean here in line 395 is that for any given tropical rational map viewed as a function from its input domain to its output domain multiple other distinct algebraic expressions represent the same function; this is precisely the problem we solve by removing redundant monomials.

---

> ### Comment · Reviewer_64tt · 2024-12-02
>
> I appreciate the authors' rebuttal. I have some follow-up comments:
> - For the reply regarding the confusing results, since the statement is about a specific trained instance of an architecture (and there is a high computational cost associated with it) I'm still not seeing what the tangible benefit is. Any actionable choice on the network would require retraining, which would render the computations moot, as they are not tied to the architecture/network class but to the specific trained instance. In what way is the count of the linear regions of the trained network informative?
> - Regarding the reply to Question 3, is the permutation group the only one that is applicable with the authors' method? Section 3.1 implies a general approach working with arbitrary discrete groups. Following up on the sample efficiency, since the authors explicitly mention that they consider the actions to be on sets rather than vector spaces, the original comment I made still stands, as $[-R, R]$ and $[-R, R]\cap \Delta$ are both infinite sets for any discrete group. To be as precise as possible, I was expecting numerical results that were supporting the factorial improvement, since that's the context it was implied in.
>
>
> I am aware of the tardiness in submitting my response and I apologize, I don't expect the authors to reply  in such a short timeframe before the deadline. I'm still hesitant about contributions and will maintain my score.

---

> > ### Author Response · Authors · 2024-12-03
> > **Response to follow-up comments**
> >
> > ## Comment 1
> >
> > There are many tangible benefits for computing the number of linear regions of a trained neural network, and potentially more provided by our exact geometric characterization of them. For instance, most theoretical results give the maximum number of linear regions a class of neural networks can generate, however, understanding how many of those are realized in practice is useful for informing their practical implementation. Moreover, the number of linear regions is a proxy for the complexity of the function represented by the neural network which can be indicative of network properties such as robustness and generalization ("Deep Networks Always Grok and Here is Why"). With our tools we can go further an determine the volumes of the regions which may provide a more accurate interpretation of this proxy. Although our tools are currently not at the scale to perform this analysis on larger networks, we can initiate the study on smaller networks and gain an intuition of how this novel geometric perspective on linear regions relates to the behavior of the neural network. The empirical realization of other measures of expressivity has also proved to be beneficial in understanding neural network performance, "On the Expressive Power of Deep Neural Networks" and "Deep ReLU Networks Have Surprisingly Few Activation Patterns".
> >
> > ## Comment 2
> >
> > Our method is agnostic to the discrete group being considered and the numerical sampling method employed; we chose the permutation group to demonstrate an application of our method in practice. As mentioned by the reviewer, the sampling domains are still infinite sets, however, note that the cube $[-R,R]$ grows exponentially as the input dimension increases, whereas $[-R,R]\cap\Delta$ does not. Thus with our method, we can maintain a certain density of points in our sampling domain while mitigating the consequences of the curse of dimensionality. Of course, this comes at a cost: our methods only provide an upper bound on the number of linear regions. However, we show in Figure 6 (of the revision) that this error is relatively low compared to the significant improvements in computation times.

---

### Official Review · Reviewer_gsPs · 2024-11-04

**Soundness:** 3
**Presentation:** 3
**Contribution:** 3
**Rating:** 6
**Confidence:** 2

**Summary:**

This paper presents a novel approach to investigating the expressivity of neural networks through the framework of tropical geometry, a branch of algebraic geometry that employs combinatorial and polyhedral methods, with key contributions including a new framework for selecting bounded sampling domains using the Hoffman constant, a method for enhancing computational efficiency through symmetry in network architecture, and a Julia library integrated into the OSCAR system for symbolic representation and analysis of neural networks using tropical techniques. The paper builds on foundational work that connects tropical rational maps with feedforward neural networks, providing theoretical insights supported by experiments that demonstrate its applicability across various network architectures. Overall, I believe this paper makes a significant contribution to the theoretical foundations of deep learning.

**Strengths:**

**Clarity:** The paper is well-organized, exhibiting a clear and logical progression from the theoretical background to the main results and their implications. Each section is thoughtfully introduced with specific motivations that enhance the reader's understanding. Notably, the three contributions are effectively aligned with their corresponding sections, reinforcing the connections among the presented ideas.

**Novelty and Significance:** The innovative application of tropical geometry to evaluate the expressivity of neural networks introduces valuable insights. By translating neural networks into tropical representations, the authors utilize tools from an underexplored area of deep learning theory. Their approach, which provides an exact count of linear regions and introduces monomial complexity as a measure of expressivity, has the potential to inspire future research on model architecture and optimization.


**Practical Implementation:** The integration into OSCAR allows for symbolic manipulation of neural networks, which has practical implications for further research and applications.

**Weaknesses:**

**Limited Empirical Validation:** The experimental demonstrations, while useful as proofs of concept, are mainly limited to small-scale networks.

**Context within Existing Methods:** A more in-depth discussion on how this tropical approach compares to or enhances traditional methods of region counting and expressivity analysis would provide valuable context.

**Notation Clarity:** Some notations in the main paper are not defined in a timely manner. For example, $P(A,b)$ is defined in Line 673 of the appendix but is used in Line 146 of Section 2 without prior explanation.

**Questions:**

See Weakness.

---

> ### Author Response · Authors · 2024-12-03
> **Response to Reviewer Comments**
>
> We thank the reviewer for their careful reading of our work and would like to respond to the concerns raised and respond to the weaknesses raised and their questions.
>
> ## Limited Empirical Validation
>
> We thank the reviewer for pointing out the limitation and we acknowledge that our experiments primarily focused on small-scale networks to facilitate conceptual analysis. That being said, the code we developed is capable of handling higher-dimensional cases. For instance, it can process networks with input dimensions of $28\times 28$ (MNIST scale) in approximately one minute. This demonstrates that our approach is inherently scalable, even though the illustrative experiments might have suggested otherwise.
>
> ## Context Within Existing Methods
>
> We appreciate the reviewer’s suggestion. To the best of our knowledge, our method is the first to count the exact number of linear regions and use tropical monomials as a measure of expressivity. Other methods in the literature only give bounds and estimates of the linear regions. We will find a suitable comparison of our method and traditional estimating methods in future work.
>
> ## Notation Clarity
>
> We apologize for the confusion. We will add pointers to the appendix for pieces of notation that are not defined in the main text.

---

### Meta-Review · Area_Chair_8nHk · 2024-12-20

**Metareview:**

The authors studied characterizations of a neural network's piecewise linear regions via tropical geometry. In particular, the authors found a ball that intersects all linear regions, and characterized the radius in terms of the Hoffman constant. Furthermore, the authors introduced a software library for the computational aspects of this research.

There are several concerns, and many reviewers asked about the Hoffman constant and how it is NP-hard to compute. Broadly speaking, there are questions about how will this characterization be useful. For example, reviewer 5XwZ raised the concern about how the number of linear regions is related to complexity measures of neural networks.

Given the discussion and the current scores, I will recommend reject.

**Additional Comments On Reviewer Discussion:**

As mentioned above, many reviewers raised concerns regarding the Hoffman constant, and how the results can be used in the future. The discussion and the lack of resolution of these issues helped confirm my view on this paper as well.

---

### Decision · Program_Chairs · 2025-01-22

Reject